# Development of a machine learning model for river bedload

Hossein Hosseiny[1], Claire C. Masteller[1], Jedidiah E. Dale[1], and Colin B. Phillips[2]

[1]Department of Earth and Planetary Sciences, Washington University in St. Louis, St. Louis, MO, 63130, USA
[2]Department of Civil and Environmental Engineering, Utah State University, Logan, UT, USA.

*Correspondence to: Claire Masteller (cmasteller@gmail.com)*

**Abstract.** Prediction of bedload sediment transport rates in rivers is a notoriously difficult problem due to inherent variability in river hydraulics and channel morphology. Machine learning offers a compelling approach to leverage the growing wealth of bedload transport observations towards the development of a data driven predictive model. We present an artificial neural network (ANN) model for predicting bedload transport rates informed by 8,117 measurements from 134 rivers. Inputs to the
model were river discharge, flow width, bed slope, and four bed surface sediment sizes. A sensitivity analysis showed that all inputs to the ANN model contributed to a reasonable estimate of bedload flux. At individual sites, the ANN model was able to reproduce observed sediment rating curves with a variety of shapes without site-specific calibration. This ANN model has the potential to be broadly applied to predict bedload fluxes based on discharge and reach properties alone.

## 1 Introduction

Bedload transport in rivers is a stochastic (Ancey, 2010; Paintal, 1971), nonlinear (Meyer-Peter & Müller, 1948; Wong & Parker, 2006), phenomenon with high dimensionality (Goldstein et al., 2019). Further, direct measurements of bedload transport are often challenging to collect reliably, especially for large, rare floods or over long periods of time. In lieu of continuous measurement, accurate estimation of bedload transport rates with minimal site-specific calibration has a number applications (Wilcock, 2001), including by not limited to quantifying channel conveyance (Slater and Singer, 2013), informing
river restoration efforts ((East et al., 2015; Warrick et al., 2015)), and approximating bedrock incision rates (Beer & Turowski, 2021). As such, there has been a long legacy of scientific inquiry towards accurate quantitative prediction of bedload transport rates, beginning in the early 1900s (Gilbert, 1914) and continuing to today (Einstein, 1937; Wilcock and Crowe, 2003; Lajeunesse et al., 2010; and recently Zhao and Nepf, 2021 among many others). A number of models of fluvial sediment transport have been developed based on semi-empirical regressions fit to flume (Meyer-Peter & Müller, 1948; Wong & Parker,
2006) and field (Recking, 2010, 2013b; Rickenmann, 1991) data, probabilistic approaches (Einstein, 1950; Furbish et al., 2012), and physics-based models (Lajeunesse et al., 2010; Parker, 1990; Wilcock & Crowe, 2003). Multi-model comparisons demonstrate that few models consistently perform well for large, multi-region datasets in part due to limitations in addressing site specific variability or due to temporal and spatial averaging (Barry et al., 2008; Gomez & Church, 1989; Recking, 2010, 2013a). As such, existing bedload flux models are not versatile enough to be applied across the range of observed river reaches
without extensive regional or site specific calibration (Goldstein et al., 2019; Kitsikoudis et al., 2015). Thus, predicting rates

of bedload sediment transport remains a persistent challenge, with predictions within an order of magnitude of direct measurements generally considered reasonable model performance (Recking, 2013a; Recking et al., 2012).

This inherent variability in bedload transport observations and the associated need for site-specific calibration efforts, has led to recent suggestions that the reliable and consistent prediction of bedload transport from reach scale parameters may
be intractable (Gomez & Soar, 2022). Indeed, there are a number of factors that give rise to variability into bedload transport rates across sites or through time at a single site including, but not limited to: spatial variability in both turbulent stresses and bed heterogeneity (Monsalve et al., 2016, 2017); grain protrusion, compaction, and structural arrangement of the bed (Church et al., 1998; Houssais et al., 2015; Marquis & Roy, 2012; Masteller & Finnegan, 2017; Masteller et al., 2019); intermittency in flux and sampling times (Bunte & Abt, 2005; Singh et al., 2009; Recking et al., 2012); upstream sediment supply (Recking,
2012; Singer, 2010; Gomez & Soar, 2022); and interactions between grain size fractions on the surface and within the bed (Wilcock, 1998; Ferdowsi et al., 2017).

Results from laboratory flume experiments and long-term field monitoring demonstrate that much of this variability may be collapsed or understood under controlled conditions. Grain protrusion within mixed grain size distributions can be accounted for through the use of hiding functions and relative reference critical shear stresses (Einstein, 1950; Ashida &
Michue, 1971; Parker et al., 1982; Wilcock & Crowe, 2003). The challenge of vertical sorting and differing grain sizes between the river bed surface and subsurface was circumnavigated through the development of surface-based transport relations (Parker, 1990). Even grain scale complexity in the particle shape can be unravelled by accounting for relative changes in fluid drag and friction (Deal et al., 2022). Field and laboratory experiments demonstrate that the impact of a hydrograph with floods of different magnitudes and shapes on bedload flux can be understood cumulatively and is linearly related to the integral
of the excess shear stress (Phillips & Jerolmack, 2014; Phillips et al., 2018). These selected demonstrations indicate that while there may be significant variability in raw measurements of bedload flux, this variability is not such that the development of a model which accurately captures patterns in bedload flux is intractable. Wholesale field application of a physically based model will continue to remain data limited; however, the introduction of longer-term monitoring stations indicates that a more nuanced physical model may be on the horizon (Rickenmann and McArdell, 2007; Rickenmann, 2018;  Gomez et al., 2021).

The known complexity of natural river processes combined with the amount of available bedload data across many sites and settings (Hinton et al., 2017; King et al., 2004; Recking, 2019) suggests that this process may be predictable from a data science approach (Geron, 2019). Machine learning (ML) approaches leverage available data to train computers to, through an automated process, determine the relative contribution of individual input variables to a measured output (Geron, 2019). In the learning process, the ML algorithm iteratively discovers patterns and relations within the data and uses them for future
predictions given similar input data. Many ML approaches do not consider the physics behind any specified problem directly, but excel at predicting nonlinear relationships with high dimensionality given sufficient training data (Hosseiny, 2021; Hosseiny et al., 2020). ML approaches can leverage variability aggregated from many existing datasets in order to improve site-specific bedload transport predictions across a range of fluvial environments. ML approaches have been previously exploited in a variety of geoscience problems including identifying vulnerability in Antarctica's ice sheet (Lai et al., 2020),

global-scale soil salinization predictions (Hassani et al., 2021), and landslide susceptibility mapping (Zhou et al., 2021). In particular, an Artificial Neural Network (ANN) approach may be particularly well-suited for bedload prediction. ANN is a well-tested and powerful method which, through an iterative and automated training process, determines the weighted contribution of numerous input parameters towards a specified output (Haykin, 2008). This iterative approach allows ANN to parse nonlinear relations between numerous input parameters, making it a flexible tool for solving a wide range of problems,

including optimization (Haykin, 2001) and data classification (Saravanan & Sasithra, 2014). Relevant to geoscience applications, ANNs have shown to be versatile tools towards more accurate description of rainfall-runoff processes (Hsu et al., 1995; Han et al., 2021), prediction of riverbed porosity (Bui et al., 2019), and for flood prediction (Hosseiny et al., 2020).

    Despite publicly available, high-quality observational data, the application of ML tools to sediment transport in rivers has, to our knowledge, remained limited. Kitsikoudis et al. (2015) used sediment concentration data from flume and field

studies, for sand (median grain size, $D_{50}$ = 0.062 mm-2.0 mm) bed rivers (Brownlie, 1981), to evaluate the performance of ML approaches: (a) ANN, (b) symbolic regression (SR), and (c) adaptive-network-based fuzzy inference (ANFIS) models. Their results show that models trained solely on flume data perform worse than those trained on field data with root mean squared errors (RMSE) of flume-trained predictions between 85% to 97% more than field-trained models. This study also found that the ANN model trained on field data performed best, with RMSE values 7.5% and 11.1% less than ANFIS and SR,

respectively. Aseghi and Hosseini (2020) trained an ANN using 102 measurements of discharge, velocity, water surface slopes, flow depth, and median grain size to develop a prediction model for bedload transport for a single site - the Main Red Fork River in Idaho. They found that the trained ANN captured bedload flux measurements more accurately than existing empirical equations – however, the wider applicability of the study may be limited because the ANN was trained using data from only a single site. Kitsikoudis et al. (2014) focuses on bedload transport within gravel-bed rivers, however the dataset is primarily

drawn from a limited geographic region of the United States (Idaho, King et al., 2004). These data are generally of high-quality, and while they integrate measurements from a number of rivers, they occupy only a limited portion of the gravel bed river parameter space. These rivers tend to be steeper, coarser grained, and shallower than average, limiting relations derived on these data to similar geographic locations (see Phillips & Jerolmack, 2019). Bhattacharya (2007) used an ANN approach using the Gomez and Church (1989) sediment flux database for sand and gravel ($D_{50}$ = 0.062 mm-64 mm) bed rivers in

subcritical flow to predict bedload and total load transport rates as a function of a combination of measured and derived input parameters. These input parameters include velocity, depth, particle diameter, slope, non-dimensional shear stress, critical shear stress, and stream power. They concluded that the RMSE of the model trained based on field data (407 observations) was on average 33.6% less than those derived from flume data and 16.4% to 249.6% less than several empirical and physically based models (Bhattacharya, 2007).

While these previous studies have demonstrated that ML models can improve upon existing sediment transport models, this suite of ML models have been trained with limited data (less than 500 observations) and under a relatively narrow range of the full parameter space which gravel-bed rivers occupy globally. Despite the increasing availability of bedload datasets, the application of ML in generating a versatile, data-driven model for predicting bedload transport across a wide

range of fluvial settings has not yet been investigated. To fill this gap, this paper develops a new ML model for predicting river bedload using an ANN approach and over 8,000 measurements from over 100 unique field sites. The performance of the proposed model is then shown to outperform four existing sediment transport models using only the publicly available data (Einstein, 1950; Recking, 2013b; Wilcock & Crowe, 2003; Wong & Parker, 2006). We finally demonstrate the utility of a broadly-trained ANN model by producing bedload transport rating curves for discharge without the need for additional site specific calibration.

## 2 Materials and Methods

### 2.1 Data summary and preparation

We use a compilation of bedload transport rates downloaded from BedloadWeb  (http://en.bedloadweb.com), a publicly available online platform that hosts both previously published field and laboratory bedload datasets compiled from scientific literature or official reports and databases (Recking, 2019). Our study focused on field-collected datasets only, as these cover a greater range of variability in terms of the key variables associated with bedload transport (e.g. discharge, grain size, slope). The database includes 10,056 individual measurements of bedload transport from more than 134 unique field sites across the globe. Each reported bedload transport data point, $q_s$ (g/s/m), in our study has an associated measurement of river discharge, $Q$ (m$^3$/s), bed slope, $S$ (m/m), flow width, $W$ (m), and the 16$^{th}$, 50$^{th}$, 84$^{th}$, and 90$^{th}$ percentiles of the bed surface grain size distribution ($D_{16}$, $D_{50}$, $D_{84}$, $D_{90}$). An additional advantage of these specific input parameters is that the static parameters (slope and grain size) can be directly measured between transport events and used to predict sediment flux from available hydrograph data (discharge and width). An important advantage of using a multi-site dataset, such as the BedloadWeb database, for model training is to encompass a broader parameter space than would be present at any individual river location. These data span a wide range of bed slopes (0.018-0.136 m/m), widths (0.3-306 m), grain sizes (D$_{50}$ 0.00013-0.22 m), and discharges (0.00005-427.5 m3/s). As such, an ANN model trained using this dataset will have significantly wider applicability than one trained on a dataset covering a smaller range. Within this database, slope and grain size are largely static variables for each site describing the river reach, while flow width and discharge are dynamic and vary in time at each site. Grain size data is a mixture of direct measurements and interpolated data under the assumption that bed surface sizes are log-normally distributed. Interpolated data is used in cases where specific percentiles of the grain size distribution have not been directly measured or reported by the original studies. In our compiled database, we used measured grain sizes whenever they are available. In five cases, $D_{16}$ values are not reported, and interpolated data is used (Recking, 2019). In 53 cases, both $D_{16}$ and $D_{90}$ were not reported, and similarly, in these instances interpolated values were used as input parameters. In the initial training of the ANN, all reported variables are used as input parameters to train the model and predict $q_s$, as we expect a model informed by all available parameters (knowledge) will have the strongest predictive power (Haykin, 2008).

Prior to model training, the data were inspected for overall quality and outliers were removed. The presence of extreme values and outliers generally degrades the overall performance of the resulting model (Geron, 2019). As such,

following procedures used in prior studies, we chose to first remove transport measurements with associated discharge values exceeding the 95th percentile (Dovoedo & Chakraborti, 2013; Kennedy et al., 1992) of all reported discharges in the database ($Q > 430$ m$^3$/s; a total of 504 points), followed by removing extreme $q_s$ values above the 95th percentile of the remaining data ($q_s > 401.4$ g/s/m; 478 datapoints) as well as those below the 10th percentile (Kennedy et al., 1992) of remaining data ($q_s < 0.1$ g/s/m; 957 datapoints). Following removal of these points, the total sample number was reduced from 10,056 to 8,117 measurements across 134 rivers. This screening process did not eliminate any individual site from the database, such that neither large nor small rivers are selectively removed during this data preparation step. While this removal of more extreme values is an important step to ensure model quality, we acknowledge that this step preferentially removes the most extreme flow and sediment transport events from the dataset. While there is significant interest in predicting sediment transport rates for extreme flow events, these largest events are the least frequently occurring in the dataset and more data would be needed to train an ANN model to reliably predict bedload flux under these conditions. Following this screening, we maintain 134 distinct datasets, emphasizing that the training data do encompass more frequently occurring small and intermediate floods across all available sites in the database. Thus, while the trained model presented here may not be appropriate to predict bedload flux in response to exceptional events in larger rivers, it can still be applied over many orders of magnitude of discharge, as described above. Following this screening process, the median number of samples across all sites is n=50. For larger rivers with maximum discharges exceeding 300 m$^3$/s (n=17), the median number of samples is reduced, n=23. However, five of these largest rivers have sample sizes exceeding the median sample size n=50, with a maximum sample size of n=146 for the Mondego River (1.8% of the full database). Thus, following the screening process, large rivers remain adequately represented in the training dataset. The 25th percentile for sample size is n=18 and the 75th percentile is n=83, with 82% of the sample sizes within one order of magnitude. Only 22 sites have more than 100 samples. The largest dataset is from Goodwin Creek, which has 307 samples and comprises <4% of the full database. Given this, we do not expect that any individual dataset should overly bias model training. Data were then log-transformed (base 10) such that each parameter distribution would more closely follow a normal distribution (see Supporting Information). Data were then scaled by minimum and maximum measurement values, such that the transformed range of values for each variable ranged from 0 to 1 (Geron, 2019; Haykin, 2008). Data were shuffled and randomly divided into two populations: a training population (80%) and a test population (20%) with equivalent distributions consistent with the full dataset.

## 2.2 Machine learning structure and implementation

Following previous applications of ML to sediment transport (e.g. (Bhattacharya et al., 2007; Goldstein et al., 2019; Kitsikoudis et al., 2015), we employ an artificial neural network (ANN) approach. The ANN framework is based on a network of connected units (neurons), most commonly comprised of single input and output layers, multiple hidden layers, where each layer contains a set of neurons (Geron, 2019; Haykin, 2008) (Fig. 1a). The ANN presented here was developed using Keras (Chollet & others, 2015), an Application Programming Interface in the Python programming language. The structure of the ANN was informed by available bedload transport data and associated measurements of discharge, channel morphology (slope

and width), and grain size (4 measurements). The input and output layers of the ANN were set to seven ($Q$, $S$, $W$, $D_{16}$, $D_{50}$, $D_{84}$, $D_{95}$) and one ($q_s$), respectively. The functions that guide the model in identifying nonlinear relations (activation functions), were set to the Rectified Linear Unit (ReLU), except one function associated with the output layer, which was set to be a sigmoid function. The ReLU($x$) returns the maximum of (0, $x$) and sigmoid($x$) returns $1/(1+\exp(-x))$. To avoid overfitting in the training process, each input segment was normalized (batch normalization) and a subset of the neurons in each layer were temporarily ignored (dropout) to add additional noise to data (Geron, 2019). The training process of the ANN model uses 80% of the bedload transport data to determine the weight coefficients of the neurons' connections that minimize prediction error. During each iteration of the ANN during the training process, the Mean Standard Error (MSE) is computed between the model-predicted data and the observational training data (Fig. 1B). We select MSE over Root Mean Square Error (RMSE) because it more heavily penalizes larger errors compared to RMSE, which is the square root of MSE, or the coefficient of determination ($R^2$). This penalization of large errors by MSE is particularly helpful in the efficient optimization of the ANN across multiple training epochs. To assess whether the model may be over-fit to the training data, we also perform a validation test of the model at every iteration of the ANN. The validation of the ML model in each iteration (epoch) was carried out by calculating the MSE on a random subset of the training dataset that is not used in that epoch. For this application, 10% of the training dataset was used within the epoch model validation step. Once the MSE of the training dataset has reached a stable minimum across many iterations (Fig. 1B), and the MSE on the validation data is consistent with this minimum, we consider the model to be sufficiently trained.

## 2.3 Comparison of ANN performance with previous bedload models

We selected four bedload transport models with varying approaches and degrees of complexity to compare to and build intuition for the predictions of the ANN model. We selected: (1) a probabilistic model developed by Einstein (1950), (2) a physics-based model developed by Wilcock-Crowe (2003), and (3, 4) two empirical models from Wong-Parker (2006) and Recking (2013). We acknowledge that these physics-based bedload transport equations could likely be calibrated to fit the available data as many of the equation coefficients are in practice tuneable to the data at hand. However, the need for site-specific sediment flux measurements to calibrate a relation severely limits the application of these bedload transport equations to most natural settings as the accurate measurement of bedload transport remains a challenging and time-consuming endeavour. Given that the aim of this contribution is to develop a predictive model that does not require any site-specific calibration, we do not undertake any additional tuning of the existing equations for bedload transport across sites prior to comparison with the ANN predictions. Within this analysis, the purpose of utilizing these four different bedload transport equations is to provide a comparison with and build intuition for the ANN approach.

We compared bedload flux measurements to predictions from these four bedload transport models and the trained ANN model (Fig. 2). All predictions were made using the 20% of data excluded from the ANN training process (test data, $n$ = 1,624). The ANN model utilizes all available data from the bedload database (7 inputs), while the bedload transport models have varying degrees of complexity, ranging from requiring four input parameters (Einstein, 1950; Wong & Parker, 2006) to

five input parameters (Wilcock & Crowe, 2003) (see Table S3). Selected previous models are valid for sand and gravel bed rivers, and therefore, the comparison is restricted to these rivers. A further description of these models is provided in the Supporting Information.

### 2.3.1. Einstein (1950)

The Einstein model (1950) assumes that bedload flux is related to the probability of a particle being eroded as a function of changes in turbulent intensity, rather than the average fluid forces acting on the particle. As such, the model relates the probability of erosion (as a function of flow intensity) to the intensity of bedload transport (Eq. 1). This method does not require a critical shear stress for incipient motion since the movement of the grain is based on probabilistic estimates. The Einstein equation tends to perform well for estimating local bedload in large rivers with uniform sand and gravel (Garcia, 2007). The implicit form of the Einstein equation is described as

$$1 - \frac{1}{\sqrt{\pi}} \int_{-(0.413/\tau^*)-2}^{(0.413/\tau^*)-2} e^{-t^2} \, dt \; = \; \frac{43.5 \, q^*}{1 + 43.5 \, q^*} \tag{1}$$

where $\tau^*$ is the dimensionless shear stress for uniform flow (Shields stress), $t$ is the integral parameter, and $q^*$ is the dimensionless bedload transport rate (or Einstein bedload number).

### 2.3.2. Wong and –Parker (2006)

Wong and Parker (2006) reanalysed the data used to develop the foundational Meyer-Peter and Muller (MPM) equation (Meyer-Peter & Müller, 1948) and found a better fit to data resulting in the following equation:

$$q^* = 3.97(\tau^* - \tau_c^*)^{3/2} \tag{2}$$

where the exponent is fixed at 3/2 and $\tau_{*c}$=0.0495 is the dimensionless threshold of sediment entrainment. The MPM equation is similar in form, but tends to overpredict bedload at higher discharges (Barry et al., 2004). Experimentally, bedload flux is well-described by Eq. 5 and similar models employing excess shear stress raised to a 3/2 power (see Lajeunesse et al., 2010), however application within different rivers and flumes typically requires that both the coefficient and threshold shear stress be treated as fitting parameters (Mueller et al., 2005; Phillips & Jerolmack, 2019). Here, for the sake of comparison, we have applied this equation using fixed coefficient and thresholds as it was not possible to estimate these parameters at each site in the database. The difficulty in estimating the threshold shear stress is a significant hurdle in the application of bedload transport equations (Buffington and Montgomery, 1997, Phillips et al., 2022).

### 2.3.3. Wilcock and Crowe (2003)

Wilcock and Crowe (2003) presented a sophisticated transport model for mixed gravel and sand, based on 48 laboratory experiments with five different sediments sizes. The fractional transport discharge in this model is estimated based on a reference parameter informed by the sediment distribution of the bed surface. This model represents a major advance by

incorporating the non-linear effects of sand content on the mobility of gravel and the overall transport rate (Wilcock & Crowe, 2003). We applied this model to the available testing dataset by estimating sand fractions from sediment grain size data, followed by estimating the reference shear stress for the geometric mean grain size. More information about this method and the steps undertaken in this study is presented in the supporting document (Text S1).

### 2.3.4. Recking (2013)

The Recking (2013b) model is a single continuous function from two equations previously developed in Recking (2010). The model can be used for sand and gravel mixtures and was developed based on 6,319 field observations and 1,317 flume measurements (Recking, 2010). The model considers sediment mobility based on $D_{84}$, as this size was observed to impact bed material mobility, flow resistance, hiding, surface armoring, and bed shear stress (Recking, 2013). The critical mobility parameter ($\tau_{*c}$) is set to a constant for sand, and as a function of the ratio of $D_{84}/D_{50}$ and the river slope, $S$.

### 2.3.5 Quantitative comparison of ANN performance and bedload models

In order to evaluate the performance of the ANN relative to these existing models, we calculated MAE for the four previous bedload transport models and the ANN model based on the direct measurements of bedload flux from the BedloadWeb database within the portion of the dataset reserved for the test ($n = 1,624$). MAE is calculated as:

$$MAE = \frac{\sum |observed - predicted|}{number\ of\ samples}. \quad (3)$$

We selected MAE as the primary criteria to assess the average model performance because it is less sensitive to extreme values (Willmott & Matsuura, 2005). To better compare under and overprediction of each model across multiple orders of magnitude, we log-transformed all bedload transport observations and predictions. This is because, based on Eq. 3, predicted values that fall multiple orders of magnitude below observed values will result in very small differences between predicted and observed, which, result, by definition, in very small MAE values. In extreme cases, MAE values computed for models that, on average, underpredict the observed data by multiple orders of magnitude (e.g. Fig. 2A) can be less than MAE values for models that equally over- and underpredict the observed data within the same order of magnitude (e.g. Fig. 2d). In this case, computing MAE on log-transformed observations and model predictions more equally weights underpredictions of each model relative to model overpredictions. Further, given that the observations of bedload transport span four orders of magnitude and are not normally distributed, this procedure helps to more equally account for model errors across the full range of the dataset.

### 3 Results

### 3.1 Model training

We found that five hidden layers, each with 600 neurons, could adequately reflect dataset measurements with minimum error (Fig. 1b). The fine-tuning of the ANN model showed that the optimum model had a batch size of 1200, a learning rate of 0.6, a dropout rate of 0.1, incorporated the mean squared error (MSE) as a loss function, and an 'Adadelta' optimizer for minimizing

the error in the training process (Chollet et al., 2015; Geron, 2019). The training process began with initial training and validation losses of MSE = 0.094 and MSE = 0.058 (Fig. 1b), and final values of MSE = 0.0126 and MSE = 0.013 after 600 iterations (epochs). Minimal improvements in error occurred between 300 and 600 epochs, indicating that the ANN model had captured the relationships between the inputs and output adequately, and further iteration would not improve performance. The ANN model performed similarly on the validation dataset (Fig. 1b), which reveals that overfitting is not an issue since the difference between training and validation errors is relatively constant and minimal (Geron, 2019; Haykin, 2008).

## 3.2 Model performance against observations

Following model training, the model with the weighting coefficients determined during training was applied to the remaining 20% of the dataset (test data) to independently predict bedload transport rates. The ANN prediction resulted in a very close prediction of the mean observed flux per unit width ($\bar{q}_{s\,ANN}$ = 25.6 g/s/m compared to $\bar{q}_{s\,DATA}$ = 31.6 g/s/m) (Fig. 2). We also performed a sensitivity test of the ANN model by training and testing a set of additional models in the same fashion as described for the full ANN model but removed a single input parameter each time (Fig. 1c). We also trained and tested an ANN model with three grain sizes ($D_{16}$, $D_{84}$, $D_{94}$) removed. We found that the performance of the ANN was most sensitive to removal of discharge leading to a 95% increase in model error (MSE) during training (Fig. 1c) and an associated 65% increase in model error when the trained model was applied to test data.

We compared site-specific, mean absolute error (MAE) values using site-specific ANN predictions to both the interquartile range (IQR) and the full range of observed bedload transport rates at each site (See Supplementary Information). We found that, on average, MAE values are less than both the IQR and the full range of $q_s$ values across 134 sites. We found 11 instances where MAE exceeds IQR and only one instance where MAE exceeded the full range of observed values at a site, comprising less than 10% of sites in the database. However, the median number of samples in these cases was 17, relative to a median of 50 samples across all sites. In addition to this, we looked at functional relationships between the site-specific model MAE for the test data versus the total number of samples at each site. We did this to ascertain whether the model was biased towards differences in sample size. We did not find any systematic or significant relationships between the sample size at any individual site and the computed errors between the ANN output and our test data. Because some of the input parameters to the ANN are dynamic (e.g. discharge, width), we also explored the absolute error between every individual observation in our database and the model input parameters. We find that there is no systematic or significant relationship between the absolute error across all data points and any individual input parameter. We did find that the lowest measured transport rates result in increased errors at some sites, which is consistent with most bedload flux models as bedload transport is often within the partial or intermittent transport regime very close to the threshold for motion (Wilcock & McArdell, 1997).

## 3.3 Comparison of ANN to previous bedload transport models

In direct comparison, the ANN model outperforms all four previous models, regardless of their complexity. The ANN prediction of bedload transport rates across the test data results in a MAE of 0.704, which is 2.5-16.8 times less than calculated MAE for the other considered models. In addition, the standard deviation for the test predictions by the ANN model was 48.2

g/s/m and the minimum amongst all models. Among the four previous bedload equations chosen for comparison, Recking (2013), an empirical model with five input parameters, performed markedly better than all other previous models with an MAE = 1.81 when compared to measured data (Fig. 2d). Einstein (1950), a probabilistic model with four inputs, performed substantially worse, with an MAE = 11.84 for the log-transformed bedload predictions. It is worth noting that the mean error ratio = -0.07, calculated for the Einstein (1950) model, is less than the other three existing bedload transport models (See Supplementary Table SX). This is due to tendency of the Einstein (1950) model to underpredict observed bedload transport rates relative to the other models. Einstein (1950) underpredicts measured bedload transport rates for more than 82% of observations, often by multiple orders of magnitude, resulting in the largest MAE when calculated using the log-transformed data (Fig. 2a). In contrast, bedload flux predictions made using Wong and Parker (2006) and Wilcock and Crowe (2003), lead to considerable overpredictions in bedload flux across sites (Fig. 2b-c). Wong and Parker (2006) resulted in an average $q_s$ = 855.7 g/s/m with a standard deviation of 2,318 g/s/m and a mean error ratio = 202.52. Wilcock and Crowe (2003) resulted in average $q_s$ = 13,278.45 g/s/m with a standard deviation of 24,011.43 g/s/m and the maximum calculated error ratios across all models, with a mean error ratio = 5,555.7. The model generally overpredicts the observed data, with the 25[th] percentile of the estimated values for the test data is 1,294-fold larger than reported measurements. In addition, high positive skewness in the predictions (skewness = 4.57) by Wilcock and Crowe (2003) showed that without independent calibration the model could not reflect the distribution of the measured data. However, MAE calculations on the log-transformed results from Wong and Parker (2006) and Wilcock and Crowe (2003) yield MAE = 2.23 and 6.59 respectively, demonstrating that while these uncalibrated models may lead to overprediction, the scale of these overpredictions is multiple orders of magnitude less than the potential underprediction of the uncalibrated Einstein (1950) approach.

We find that, without site-specific calibration, the trained ANN developed in this contribution most reliably reflects the distribution of the measured bedload data in the training dataset. Of the uncalibrated existing bedload transport models, the approach of Recking (2013) most reliably reflects the measured test data.

**4 Discussion**

We demonstrate that the trained ANN model provides a robust prediction of available test data. This is particularly encouraging because the model is trained using a dataset with wide parameter ranges compiled from many sites across the world, suggesting that it may be readily applied to any site which falls within the existing distributions of the training dataset with fairly good results (see Supplementary Information). Caution should be applied in the application of this ANN for input parameters outside of the parameter distributions for which it was trained. Admittedly, the ANN model leverages all seven available inputs from the BedloadWeb database, whereas previous models only utilize a subset (Table S3) and as such, it is not entirely surprising that the ANN outperforms existing models. However, it is worth noting that, to our knowledge, there is no available empirical or theoretical bedload model that would similarly leverage all of these input parameters. ANN model sensitivity testing revealed that each of the seven parameters aides in the final prediction, however the removal of discharge produced the largest errors by far. This result is also unsurprising, and isconsistent with findings from other recently developed sediment transport models (e.g. Cohen et al., 2022). Bedload flux is chiefly a function of the fluid stress applied to the bed in excess of the

threshold for motion and thus primarily dependent on how channel discharge maps to stress through the channel cross section (Meyer-Peter & Müller, 1948; Wong & Parker, 2006). It is worth noting, however, that the trained ANN model which does not include discharge only has a MAE = 21.1 g/s/m compared to the full ANN MAE of 15.8 g/s/m, which is still less than those from all previous models (Table S5). It should be noted that all four existing bedload transport models require some form of discharge (or shear stress) data to make predictions. All other ANN models trained on only a subset of the input parameters showed an increase in model error (MSE) in the test phase of up to 12% relative to the full ANN model. Across these sensitivity runs ANN model error was most sensitive to the removal of channel width (MSE increase of 12%) and least sensitive to the removal of $D_{90}$ (MSE increase of 0.8%). These findings are consistent with those from a sensitivity analysis of the global-scale model WBMSed (Cohen et al. (2022) and recent sediment transport models developed using a stream power approach (Lammers and Bledsoe, 2018). Across all cases, increases in total error of this class of ANN models (average MSE = 1546.0 $g^2/s^2/m^2$) is still significantly less than the four uncalibrated bedload models (minimum MSE = 6215.1 $g^2/s^2/m^2$).

We suggest that the relative insensitivity of ANN performance reflects the inherent self-organization of alluvial river systems (Leopold et al., 1960; Gary Parker, 1978; Phillips & Jerolmack, 2016). Alluvial rivers evolve towards a stable geometry that reflects a condition at which the bankfull flood will only slightly exceed the threshold for motion and initiate bedload transport (Dunne & Jerolmack, 2020; Parker, 1990). By extension, if a river is at or near this stable state, its width, slope, and surface grain size distribution, all hold information about channel size and therefore discharge required to transport sediment. We suggest that the machine learning approach, which incorporates all these inputs, better captures the covariation between channel characteristics and their influence on bedload transport rates in natural systems when compared to more deterministic models. This is, in part, due to the model training, which is explicitly aimed at parsing the functional relationships between these covaried input parameters.

The robust performance of the trained ANN across many sites also demonstrates that potential sources of variability may be absent in a particular site and that the ANN successfully captures an expected average behavior. Alternatively, these effects may be embedded within correlations between model input parameters. For example, it has been demonstrated experimentally that decreased sediment supply can result in coarsening of the bed surface (Dietrich et al., 1989). Thus, the effect of relative differences in sediment supply may be implicitly accounted for in the ANN results due to differences in the grain size input parameters relative to channel width and slope measurements. If so, this only reinforces the critical importance of river self-organization in setting bedload transport rates (Phillips & Jerolmack, 2019) and the ability of the ANN to parse this organization through a data-driven approach. The ANN cannot explicitly define the sources of potential variability given the available input parameters, but this is also beyond the scope of this contribution.

Inspection of the model predictions (Fig. 3) shows that the Wong and Parker (2004) and Wilcock and Crowe (2003) models tend to overpredict observed fluxes, but generally, capture the correct shape of the observed data and therefore could likely be calibrated to match the observed data. Calibration of bedload transport functions through adjustments to the leading coefficient and/or the threshold term can generally increase their utility (Hinton et al., 2017). However, these calibration parameters are not always easy to estimate and usually require direct measurements of bedload flux. Phillips and Jerolmack

(2019) specifically analysed field sites to investigate channel geometry and the threshold of motion and were only able to reliably calibrate bedload functions for 68 of 132 sites (51.5%). Application of empirical functions can require additional derived or calculated parameters such as shear stress. Shear stress is not necessarily challenging to derive by assuming steady, uniform flow; however, even shear stress data is rarely available at the majority of stream monitoring sites and can require a complicated set of processing routines for gaged sites (see Phillips and Jerolmack, 2016). More notably, the generally poor

predictions from the physically-based and semi-empirical bedload transport models (Fig. 2) highlights the challenge in utilizing any bedload transport equation to predict or construct a rating curve without existing site-specific flux measurements. A primary advantage of this ANN model is that it utilizes either parameters that are directly and consistently measured at stream gages (flow), measured from high-resolution topography (slope, width), or can be measured during low or no flow periods (grain size). For the majority of sites, both slope and grain size are static site variables and this presents a major advantage of

this ANN model for predicting bedload transport at gaged sites where direct measurements of bedload are not available to develop empirical rating curves or to calibrate other existing bedload functions.

One application of the ANN model developed here is to construct bedload transport rating curves for a broad range of gaged rivers. We selected a small subset of rivers that cover a wide range of parameters from the dataset used in this study to highlight the ANN model output (Fig. 3). These simple results highlight how the ANN approach can be used for the

prediction of bedload transport at gaged sites without additional site-specific calibration. The strength of the ANN model should allow for this approach to be relatively easily adapted to any gaged catchment with similar parameters or site without prior transport measurements to estimate bedload flux based on a hydrograph and reach scale estimates of bed grain size and slope. Within the US Geological Survey National Water Information System, there are thousands of potential gages. Furthermore, this model could be paired with spatially distributed hydrologic models if sufficient grain size measurements

could be made and could also be readily applied within global-scale sediment flux models (such as WBMSed, see Cohen et al., 2022) or in Earth System Models (e.g. Tan et al., 2021; Li et al., 2021) where additional necessary parameters can be modelled or estimated from global compilations (Tan et al., 2021; Li et al., 2021; Cohen et al., 2022).

**5 Conclusions**

This paper presented an artificial neural network (ANN) model for predicting river bedload. To do that, a large, measured

bedload dataset, including 8,117 data points from 134 rivers, was gathered from the BedloadWeb, a free public online platform. The structure of the ANN included an input layer, an output layer, and five hidden layers with 600 neurons. The inputs to the model included temporally variable river discharge and flow width, and static measurements of bed slope and grain size (specifically $D_{16}$, $D_{50}$, $D_{84}$, and $D_{90}$). A sensitivity analysis was carried out to show the sensitivity of the model with the input parameters. The results showed that the ANN model was most sensitive to the river discharge and least sensitive to the largest

grain size ($D_{90}$). Our analysis suggests that including all available parameters in the ANN model better captures the covariations between the input and output parameters. Further, the ANN model provides robust prediction of the test (unseen) bedload data ($n = 1,624$) within the bounds of one order of magnitude. We highlight that an advantage of this ANN model is that it was developed on a broad range of rivers and appears to accurately capture the variation in the data, making this model a good

candidate for predicting bedload fluxes at gaged sites. The proposed machine learning model in this research lays the
foundations for efficient and accurate predictions of river bedload within the broadest array of rivers to date.

**Data availability**

Original bedload data sets are available at both http://en.bedloadweb.com. Input data as used in this contribution are also published on Zenodo under a GNU General Public License at https://zenodo.org/record/7641313#.Y-vfbezMIeY, in additional
to the trained ANN model, and all associated model output described in this manuscript.

**Supplement**

The Supporting Information for this contribution provides additional details on the methods used for calculation of bedload transport rates with previous models, as well as additional summary statistics associated with the original datasets and model-
predicted values.  All data associated with this manuscript, the trained ANN model, and a sample Jupyter notebook for model implementation are published on Zenodo under a GNU General Public License at https://zenodo.org/record/7641313#.Y-vfbezMIeY.

**Author contributions**
HH conceptualized the research and led the processing of the data, developing machine learning algorithms, visualizations, and writing the initial draft. CM and CP developed the idea, provided feedback, and contributed to the editing and writing of the manuscript. JD assisted with additional data analysis during manuscript revision and preparation of the supplementary code. All authors were responsible for critical contributions and passing the final paper.

**Competing interests**
The contact author has declared that neither they nor their co-authors have any competing interests.

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

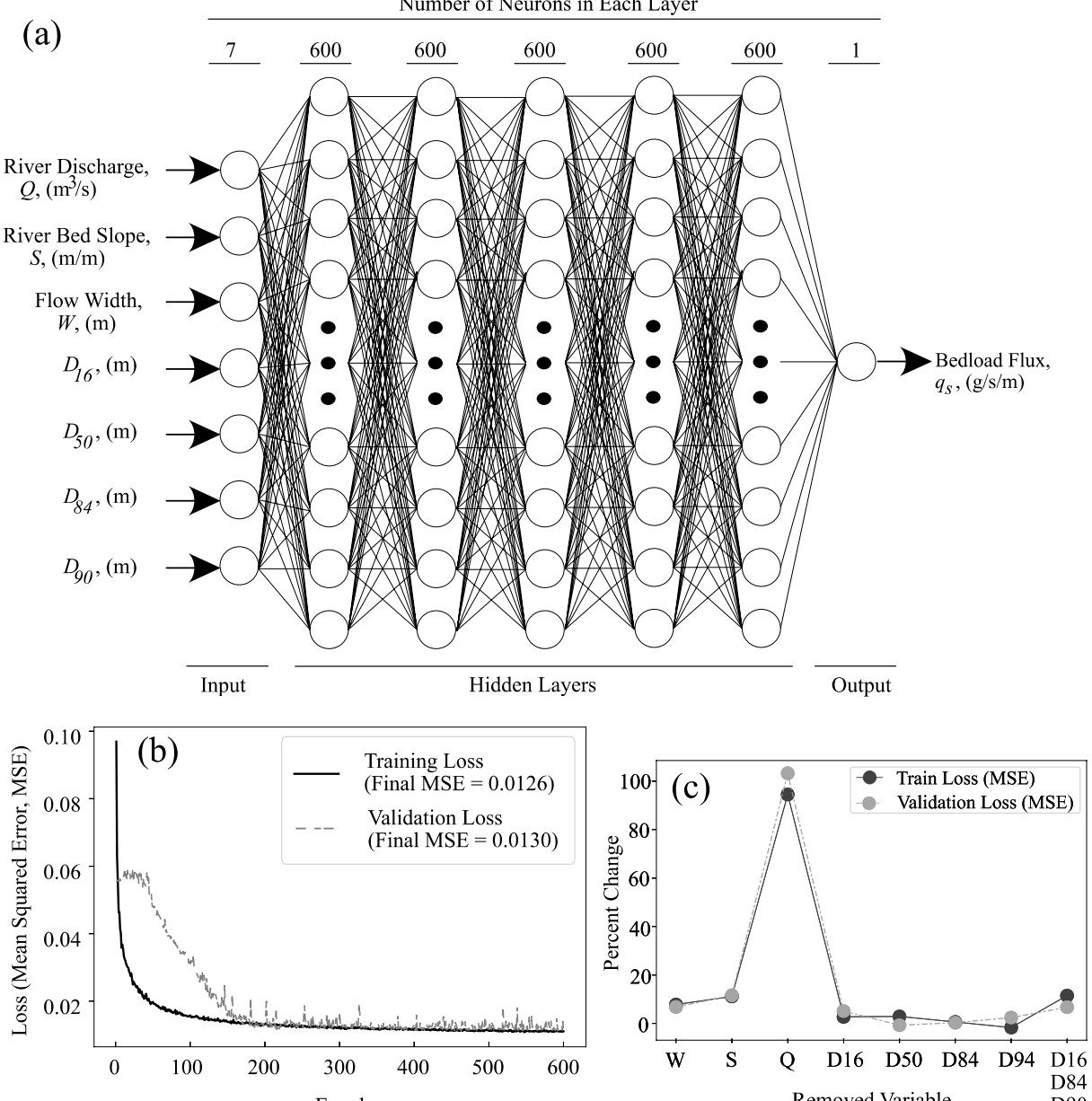

Figure 1. (a) Structure of the ANN model developed in this study with 7 input parameters. (b) Learning curves illustrate the decline in mean squared errors for training and validation. (c) Variations in ML model performance in training and validation due to changes in model input variables.

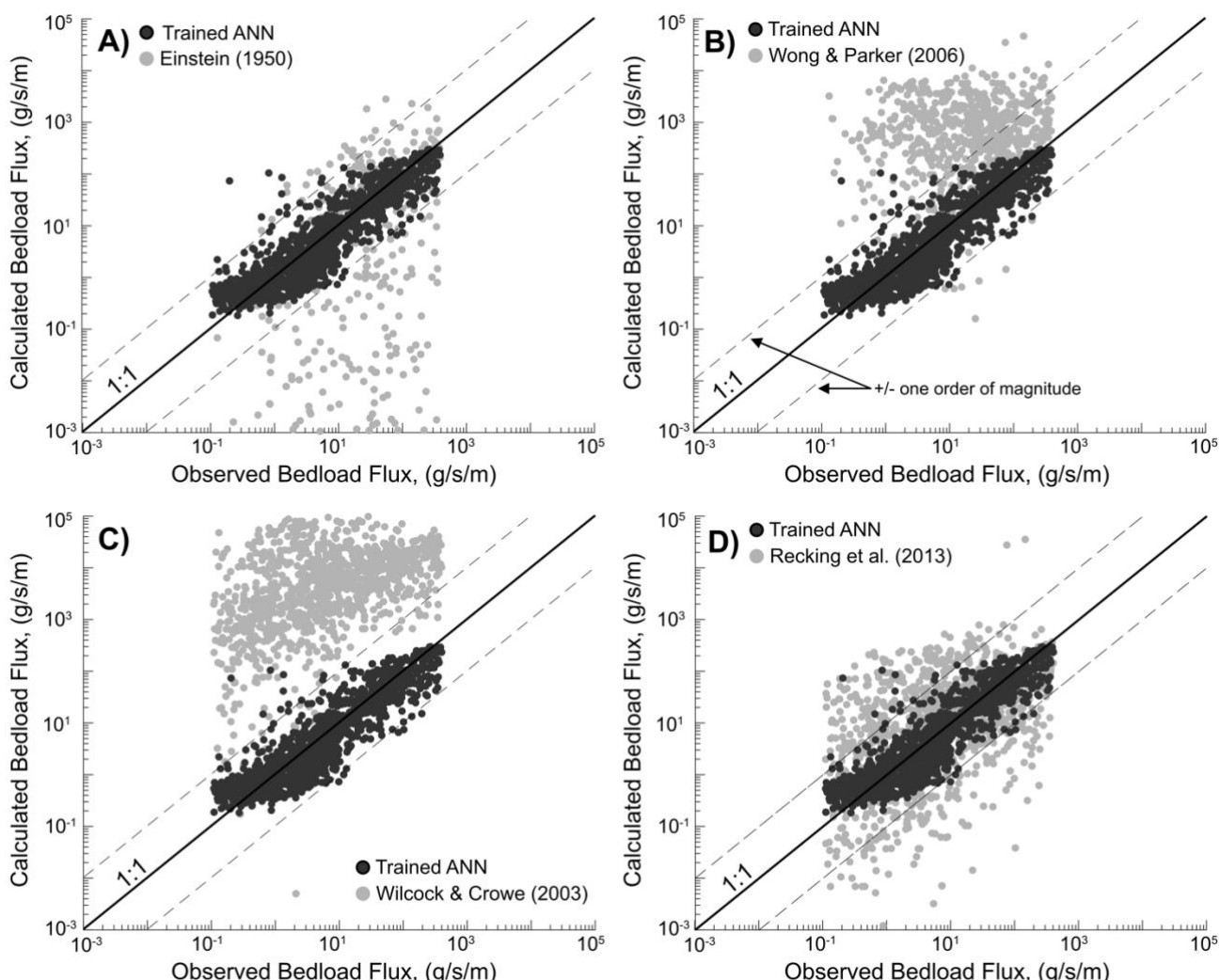

Figure 2. Comparison between ANN prediction for the test data (gravel and sand bed rivers) and previous models of (A) Einstein, (B) Wong-Parker, (C) Wilcock-Crowe, and (D) Recking (2013). Note that calculated Einstein values below 1e-3 g/s/m are not shown in the plot for legibility.

615

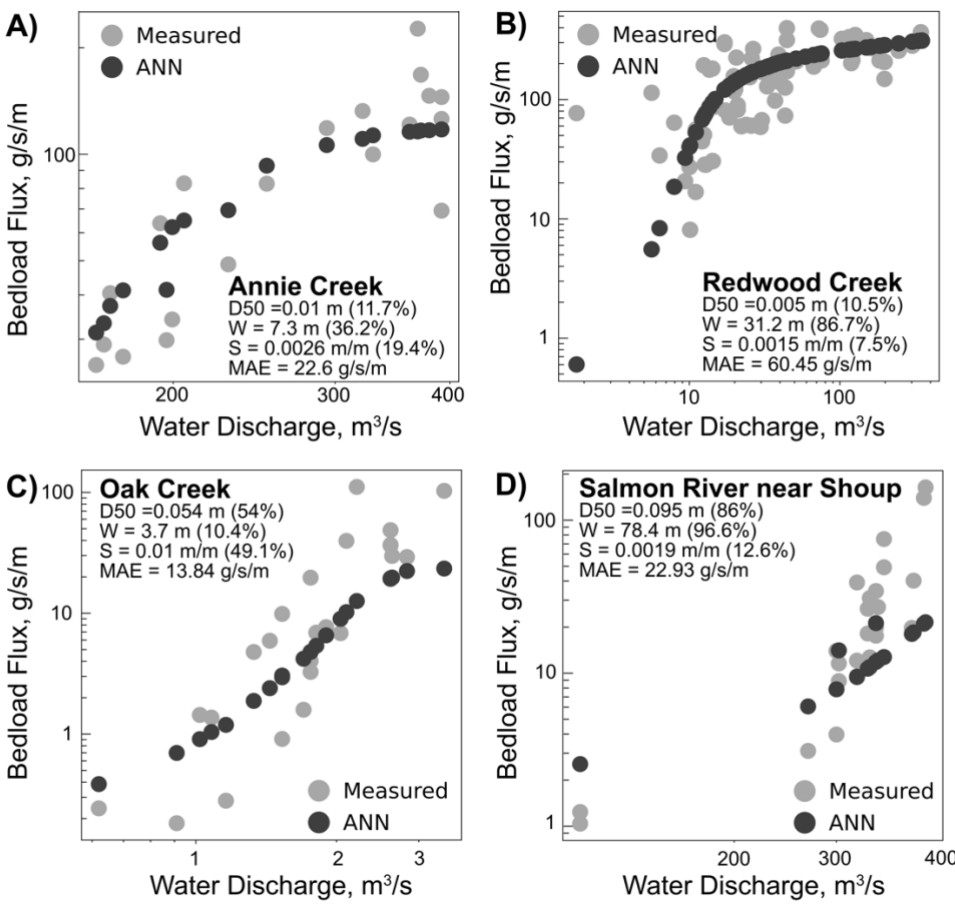

Figure 3. Example of the ANN model developed in this study applied to construct bedload transport rating curves for several sites. The numbers in parenthesis show the percentiles of each variable relative to the whole dataset.

620