# Peer review of "Development of a machine learning model for river bedload"

_Earth Surface Dynamics, 2022_

## Referee Comment (RC2)

Review of: Hosseiny *et al.* Development of a machine learning model for river bedload
https://doi.org/10.5194/esurf-2022-23 -- Basil Gomez

It has recently been claimed that the search for a formula or model that can "be broadly applied" is a fallacious pursuit, as not all rivers function in the same transport regime (Gomez and Soar, 2022). This is because, although bedload transport efficiency and the size of sediment in motion are adjusted to the environmentally controlled rate at which sediment is supplied to a river system (Gomez, 2022), the time-variations in transport rates observed at a reference section are the product of transitory, in-channel changes in sediment availability; related, for example, to the passage of bedforms, scour and fill, the formation/breakup of armor, or the injection/exhaustion of sediment derived from proximal sources.  In only one transport regime does the narrow range of inherent variability in the transport rate at a given flow magnitude indicate that it might be possible to compute temporally representative transport rates for a range of flows.  It has also been suggested the sophisticated passive and active monitoring technologies, that routinely generate long-term, continuous high-resolution datasets, have transformed bedload research from a data-scarce (theory-driven) into a data-rich (data-driven) science (Gomez *et al.*, 2022). Through the use of modern methodologies designed to tease out trends, reveal structure in bivariate (and multivariate) datasets, these 'big data can help elucidate nonlinear and complex behavior, reveal structure and trends that were previously obscured, and link bedload transport dynamics to the underlying drivers across different timescales.  A procedure (quantile LOWESS) it is advocated be applied to such data is also a machine learning task (Gomez and Soar, 2022).

Machine learning may indeed afford "a compelling approach to leverage the growing wealth of bedload transport observations towards the development of a data driven predictive model", but I fundamentally disagree with the authors' approach to predicting (or testing methods for predicting) bedload transport rates.  Machine learning techniques clearly have the potential to help isolate patterns (that must necessarily be explained in physical terms) if the data utilized embrace the specificity of rivers in the universe of fluvial systems with respect, for example, to the transport regimes or efficiency metapopulations Gomez and Soar (2022) and Gomez (2022) identified.  However, many of the rivers in the collection of data the authors draw on are referred to in the three aforementioned papers, and it should be abundantly clear that applying any machine learning technique to such a profusion of data is unlikely to yield results that "lay[s] the foundations for efficient and accurate predictions of river bedload".  This is because, as Gomez and Church (1989) demonstrated, not all bedload transport data are created equal; some characterize 'equilibrium conditions', while other data, for example, represent the condition of partial transport where not all size fractions are transport in the same proportion as they are present on the bed.

In light of the above it would seem, at the very least, appropriate to specify the transport conditions to which a particular formula pertain, because most formulae attempt to predict the maximum rate of bedload transport for a given set of conditions and use data that match these criteria in any test of their performance.  Moreover, there are also obvious errors in the data set the authors use that could easily be resolved with reference to the original works.  As an example, I offer my (Gomez, 1983) data that are incorrectly recorded as having a uniform $D_{50}$ of 11m when, in fact, reference to the original paper shows a $D_{50}$ of from ~3 mm to ~1.3 mm was specified for each measurement (transect – for reference, the size of the H-S sampler is also incorrectly specified and only an unspecified portion of the data are utilized).  Moreover, the conditions these (my) data describe are clearly not a test of any bedload transport formulae or

methodology for prediction!  There is also nonconformity in the conditions the data seek to describe, inasmuch as some of the essential variables (the energy slope is the most common example) are reported as constant values, whereas in actuality they vary.  The median grain size of the bedload is another value which is, in fact (see Gomez, 2022), rarely specified reported for each rate and is often a surrogate value (such as the median diameter of the local bed material, from which the bedload is assumed to have been derived). For all the above reasons, it would seem unlikely that any machine learning process would provide intelligible or meaningful results when applied indiscriminately to the growing wealth of bedload transport observations.

      I also have a concern about the data screening, which apparently involved arbitrarily removing extreme (the upper and lower $5^{th}$ percentiles) data, because "outliers generally degrade the overall performance of the resulting model", and the indiscriminate use of four bedload transport models to compare to and build intuition for the predictions of the ANN model.  The authors "compared bedload flux measurements to predictions from these four bedload transport models and the trained ANN model".  Do they really expect, for example, the Wilcock and Crowe or Recking models to be applicable to Mountain Creek, a sand-bed stream with an assumed bedload $D_{50}$ of 0.9 mm?

      In summary, I am sensitive to the fact that the authors may have been unaware of, or chose to ignore, papers that were published as their manuscript was being prepared for publication.  However, I cannot support the arbitrary and indiscriminate use of data and the application of bedload transport models in their study.  There are a number of very large (comprising hundreds to tens of thousands of measurements, see Gomez *et al.*, 2022 and Gomez and Soar, 2022 for examples), high quality bedload transport data sets to which the bedload transport models they utilize are eminently applicable that would also serve as a rigorous test of the ANN model they have developed.  My recommendation is that the paper be withdrawn as I do not consider it to be suitable for publication in its present form.  The ANN model, which represents the core of the paper, may have utility, but it needs more meticulous testing using the 'big data' to which it is inherently more applicable.

Gomez, B., 1983, Temporal variations in bedload transport rates: the effect of progressive bed armouring, *Earth Surface Processes and Landforms*, **8**, 41 54.

Gomez, B., 2022, The efficiency of the river machine, *Geomorphology*, **410**, 108271, doi:10.1016/j.geomorph.2022.108271.

Gomez, B. and Church, M., 1989, An assessment of bedload sediment transport formulae for gravel bed rivers, *Water Resources Research*, **25**, 1161-1186.

Gomez, B. and Soar, P.J., 2022, Bedload transport: beyond intractability, *Royal Society Open Science*, **9**: 211932, doi: 10.1098/rsos.211932.

Gomez, B., Soar, P.J. and Downs, P.W., 2022, Good vibrations: Big data impact bedload research, *Earth Surface Processes and Landforms*, **47**, 129–142.

---

## Author Comment (AC1)

Paper is interesting and well-written, but my main issue is regarding the lack of novelty. Authors just applied old model of ANN to bedload prediction. We have many new models like tree-based, rule based,... deep learning and so on. Due to lack of novelty, I have to reject the paper. Sorry for the decision.

Response:
Thank you for your feedback.

The primary objective of this contribution is to apply an established machine learning tool, ANN, to the notoriously difficult problem of accurately estimating bedload transport rates across a range of fluvial environments. It is not the intent of this manuscript to apply the latest machine learning method. In this contribution we present an ANN trained model that is able to estimate bedload flux based on a combination of static and dynamic input parameters across a wider parameter space than any previous study (see expanded discussion on this in response to Reviewer 3). We demonstrate that this trained model outperforms existing, widely-used bedload transport models as well as previous, more limited, applications of ANN for bedload estimation. We show that this approach provides a versatile model for predicting bedload for a wide range of rivers without the need for model parameter calibration at any individual site. In the revised version of this manuscript we will edit the introduction to clarify the aim and scope of the paper.

We selected the Artificial neural Network (ANN) for this research because it is well-tested a powerful tool for modeling nonlinear problems with high dimensionality (Haykin, 2008). The proposed ANN is composed of multiple layers, which, by definition, is a deep learning approach. We acknowledge the reviewer's statement that there are many machine learning algorithms available. However, we disagree that the "newness" of any particular type of ML model or data analysis technique, in and of itself, defines the overall novelty of a contribution, but rather, the novelty lies in the application of the method to a new scientific question or challenge - here being the accurate prediction of bedload transport across a broad range of settings.

Each machine learning approach has its own strengths, limitations, and constraints. We opted to apply ANN for a number of reasons. ANN is an established tool. It has been shown to be versatile, with applications in geoscience including rainfall-runoff processes (Hsu et al., 1995; Han et al., 2021), turbidity currents (Naruse & Nakao, 2021), and prediction of riverbed porosity (Bui et al., 2019). Further, previous work focused on the application of machine learning techniques to predict bedload transport found that ANN outperformed alternative ML approaches - as described in the introduction of the submission. We aim to build on this existing work, with a focus on expanding the model parameter space through the use of the bedloadweb database, as model performance is only reliable under the range of conditions for which the model was trained. This is an important point as previous bedload databases used within ML and ANN models were primarily derived from a limited geographic region which can bias the training data (See Phillips & Jerolmack, 2019). An additional advantage of ANN is that it is well-established and as such can be easily implemented by other users. We will more

clearly outline the broader applicability of ANN in geoscience, and with regard to the estimation of bedload transport rates, in the revised introduction of the paper.

It is worth pointing out that decision-tree models can be very sensitive to small variations in the training dataset (Geron, 2019), resulting in very different results due to these variations. Decision-tree approaches can also over-fit the data depending on the complexity of the classification trees. Rule-based machine learning is based on a number of if-then conditional statements and can sometimes require significant user knowledge, particularly for supervised learning routines (Nunez et al., 2006). Often, these types of approaches may use different models for prediction, based on how input data is classified by the tree structure or any identified set of relational rules. While these approaches may be of potential interest for future work in this area or to answer different questions related to bedload transport, it is beyond the scope of this contribution. Our aim is to instead use ANN to develop a singular model that can be broadly applied across the full parameter space of the training data, as this, in and of itself, represents an advance towards more accurate prediction of bedload transport.

In the revised version of the manuscript, we will more clearly articulate the reasoning for the selection of ANN compared to other ML methods. We will also discuss the potential for other types of machine learning methods to highlight different characteristics of the data in the discussion of the paper. Further, we will discuss the strengths and limitations of the development and application of a universal model for the description of bedload transport presented in this contribution.

- Haykin, S. (2008). Neural Networks and Learning Machines. In McMaster University (Third). Pearson Prentice Hall.

- Han H, Choi C, Kim J, Morrison RR, Jung J, Kim HS. Multiple-Depth Soil Moisture Estimates Using Artificial Neural Network and Long Short-Term Memory Models. Water. 2021; 13(18):2584.

- Naruse, H. and Nakao, K.: Inverse modeling of turbidity currents using an artificial neural network approach: verification for field application, Earth Surf. Dynam., 9, 1091–1109, https://doi.org/10.5194/esurf-9-1091-2021, 2021.

- Bui, Van Hieu, Minh Duc Bui, and Peter Rutschmann. 2019. "Combination of Discrete Element Method and Artificial Neural Network for Predicting Porosity of Gravel-Bed River" Water 11, no. 7: 1461.

- Phillips, C. B., and D. J. Jerolmack. "Bankfull Transport Capacity and the Threshold of Motion in Coarse-Grained Rivers." Water Resources Research 55, no. 12 (2019): 11316–30.

- Geron, A. (2019). Hands-on Machine Learning with Scikit-Learn, Keras & TensorFlow (2nd ed.). O'REILLY.

- Nunez, H., Angulo, C., & Catala, A. (2006). Rule-Based Learning Systems for Support Vector Machines. Neural Processing Letters, 2

---

## Author Comment (AC2)

Thank you for your feedback, while we do not agree with every statement we certainly appreciate the effort put into reviewing our manuscript. We have broken the comments apart below for clarity.

It has recently been claimed that the search for a formula or model that can "be broadly applied" is a fallacious pursuit, as not all rivers function in the same transport regime (Gomez and Soar, 2022).

There is a long legacy of scientific inquiry devoted to the accurate prediction of bedload transport rates, beginning in the early 1900s (Gilbert, 1914) and continuing to today (Einstein, 1937;, Wilcock & Crowe, 2003;, Lajeunesse et al., 2010; and recently Zhao & Nepf, 2021 among many others). We feel that this excerpt from Wilcock (2000) articulates the value in efficient and accurate prediction of bedload transport rates particularly well:

> "many large-scale problems in fluvial geomorphology would benefit from the availability of an efficient means of estimating sediment transport rates.... A method for estimating bed material flux that is both practical and accurate would allow the strong constraint of sediment mass conservation to be applied to a larger number of problems. The need for an efficient means of estimating sediment transport extends to applied problems."

Direct measurements of bedload transport are often challenging to collect reliably, especially for large rare floods or over long periods of time. In lieu of continuous measurement, accurate estimation of bedload transport rates with minimal site-specific calibration has a number of clear applications, including by not limited to quantifying channel conveyance, informing river restoration efforts, and approximating bedrock incision rates (Beer & Turowski, 2021). As such, we feel there is considerable scientific interest in and a strong basis for the development of a flexible model for bedload transport that can be used in scenarios where high-resolution data is unavailable.

The reviewer cites their recent paper suggesting that these endeavors may be misguided or fruitless. To us, this seems to be a philosophical difference between the reviewer and our team. We don't disagree with the reviewer that there can be considerable scatter in bedload transport data and that this scatter may arise for a number of reasons. Bedload transport data are noisy, which is precisely why prediction is challenging. However, there is also considerable evidence in the scientific literature that indicates that the prediction of bedload flux remains possible. As such, we feel there is value in the development of a model that can make these approximations, within some clearly quantified margin of error. The results of the ANN model presented in this paper demonstrates that this is achievable.

Flume experiments have demonstrated that some of the variability in observed bedload transport rates can be collapsed under more controlled conditions. For example, Deal

et al., (pre-print) demonstrated that when differences in grain shape are accounted for, flume experiments using different grains could all be collapsed onto a single bedload transport rating curve. Phillips et al., (2018) demonstrated that an impulse framework successfully collapsed total bedload flux measurements for floods of different shapes.  In field settings, there is a wealth of literature demonstrating the utility of hiding functions to better capture bedload transport of a grain size mixture (e.g. Parker et al., 1982, Recking 2010). Given these demonstrations, which represent a small number of examples in a much larger body of scientific contributions, we argue that though the data may be variable, there is considerable evidence that this noise should not preclude prediction.

We feel as though the model presented here represents a novel contribution because 1) it is data-driven in its approach, 2) it leverages datasets from many diverse sites - accounting for the inherent variability in bedload transport rates, and 3) it uses an expanded number of easily measured or estimated parameters as model inputs compared to existing models.  Our results demonstrate that the model reliably reproduces direct measurements of bedload transport on unseen data.

In a revised version of this manuscript, we will expand upon the importance of accurate bedload prediction and its applications.  We will also review the ample body of work towards accurate prediction of bedload transport and the benefits and shortcomings of these approaches more generally.  We will discuss sources of variability that can be introduced both temporally at a single site and across multiple sites that may complicate prediction.  We will also discuss past work that has demonstrated that some of this variability can be collapsed towards a generalized bedload transport relation.  We will more clearly motivate how an ANN approach can capture and reduce some of this variability without explicit consideration of every potential cause for noise/variability.

This is because, although bedload transport efficiency and the size of sediment in motion are adjusted to the environmentally controlled rate at which sediment is supplied to a river system (Gomez, 2022), the time-variations in transport rates observed at a reference section are the product of transitory, in-channel changes in sediment availability; related, for example, to the passage of bedforms, scour and fill, the formation/breakup of armor, or the injection/exhaustion of sediment derived from proximal sources. In only one transport regime does the narrow range of inherent variability in the transport rate at a given flow magnitude indicate that it might be possible to compute temporally representative transport rates for a range of flows.

We agree with the reviewer that sediment supply effects can introduce temporal variability into time series records of bedload transport rates.  Sediment supply effects may not be the only factor that introduces observed variability. For example, dynamic erosion thresholds due to subtle rearrangement of the bed surface (Masteller and Finnegan, 2017), the history of flood events (Masteller et al., 2019), and the shape of the hydrograph (Phillips et al., 2018) are additional processes that may introduce

temporal variability into measured bedload transport rates.  Beyond these, there are additional sources of variability that may arise across sites, again due not only to sediment supply effects, but also due to differences in hydroclimate, tectonics, land-use, or other local factors.  We agree that this variability is not yet explicitly captured by sediment transport models, and can present a challenge towards the accurate prediction of bedload transport rates. However, we disagree that this variability is so overwhelming as to preclude progress towards improved prediction at any individual site or across many sites.

Specifically, we would like to point out that, to our knowledge, no existing quantitative models for bedload transport explicitly account for differences in sediment supply (or any of the other effects) described above. This is precisely why we feel that a machine-learning or ANN approach is a good tool to tackle this problem. ANN can parse nonlinear relationships between many input parameters and determine their relative influence on the output. While sediment supply may not be an explicit parameter that can be directly or reliably measured, it is likely, however, that sediment supply effects are embedded in one or more of the input parameters in our trained ANN.  For example, sediment supply differences can give rise to varying levels of grain protrusion (Yager et al., 2012), which an ANN model may conceivably capture through differences in surface grain size distributions relative to channel width and slope measurements.  Thus, by leveraging all these data together as input parameters, the ANN-estimated bedload flux values may implicitly account for supply effects when making bedload transport predictions.  It is true that the ANN cannot explicitly define the sediment supply given the available input parameters, but this is well beyond the scope and goals of this contribution. We will add a short acknowledgement of potential sediment supply effects and the potential for ANN to implicitly capture these effects in the discussion.

It has also been suggested the sophisticated passive and active monitoring technologies, that routinely generate long-term, continuous high-resolution datasets, have transformed bedload research from a data-scarce (theory-driven) into a data-rich (data-driven) science (Gomez et al., 2022). Through the use of modern methodologies designed to tease out trends, reveal structure in bivariate (and multivariate) datasets, these 'big data can help elucidate nonlinear and complex behavior, reveal structure and trends that were previously obscured, and link bedload transport dynamics to the underlying drivers across different timescales.

We do not disagree. However, these monitoring techniques are not yet wide-spread or may be inappropriate to deploy in every case in which an estimate of bedload transport would be useful as they are costly and often require significant engineering efforts. There has been excellent work done in this area, and a number of high-resolution bedload transport records do exist, but many of these records are not yet publicly available.  Further, existing work focused on calibrating bedload transport models from these datasets has indicated that significant site-specific calibration is still required and the applicability of these calibrated models to other sites remains limited.

The strength of the use of the bedloadweb database over a single continuous high-resolution dataset lies in the parameter space over which the model can be trained and then applied.  The bedload.web database encompasses a range of slopes (0.018-0.136 m/m), widths (0.3-306 m), grain sizes (D50 0.0003-0.22 m), and discharges (0.00005-427.5 m3/s) that would not be present in a single location.  It is certainly possible to add continuous, high-resolution datasets for single sites to the ANN that we have developed to aid in its training and testing as long as all input variables are available.  However, because this data will likely fall within the existing distributions used to train the ANN model, we do not expect the incorporation of additional data from a single site to have drastic effects on model output across all sites. Additionally, introducing datasets with a very large number of samples compared to other datasets may lead to issues with model overfitting.

In a revised version of this contribution we will place the work within the context of these site-specific monitoring advances and the associated calibration curves that have been developed from these efforts.  We will reinforce the reasoning behind the development of this ANN and its potential application compared to these calibrated models.  We will also address the potential impact of incorporating these large datasets into model training (see expanded response to Reviewer 4).

We encourage the reviewer to incorporate their data into the efforts of larger repositories such as Bedloadweb as some of the mentioned high-resolution datasets do not appear to be publicly available as the referenced supplementary material contains dead hyperlinks.

 A procedure (quantile LOWESS) it is advocated be applied to such data is also a machine learning task (Gomez and Soar, 2022). Machine learning may indeed afford "a compelling approach to leverage the growing wealth of bedload transport observations towards the development of a data driven predictive model", but I fundamentally disagree with the authors' approach to predicting (or testing methods for predicting) bedload transport rates.

ANN is a significantly different approach from quantile LOWESS. ANN utilizes multiple input parameters to model the specified output through an iterative optimization approach.  The aim of this paper is specifically to evaluate the performance of ANN on a large database of bedload transport measurements across a wide parameter space to leverage the richness of the data available from bedloadweb.

While a quantile LOWESS does offer considerable flexibility as a fitting function to datasets with a complex structure, this approach is significantly different from the ANN approach described in our study.  For example, LOWESS requires a large, densely sampled dataset due to its reliance on the local data structure when computing a fit, whereas ANN can leverage both dense and sparse datasets in aggregate towards an optimized model. Further, the regression from LOWESS is computed between a pair of variables, limiting the datasets for which this technique can be applied and the

complexity of the model that can be derived from the data. In contrast, ANN can accept
multiple input parameters and leverage all inputs (in the case of this study, 7 in total)
towards prediction.

Our results demonstrate that ANN methods are indeed able to estimate bedload
transport rates to within an order of magnitude and with an average MAE of 16.1 g/s/m.
This performance indicates that ANN is a viable method towards a general model for
bedload transport prediction without site-specific model calibration.

In a revised version of the manuscript we will emphasize the differences between these
approaches and their associated applications.

Machine learning techniques clearly have the potential to help isolate patterns (that must
necessarily be explained in physical terms) if the data utilized embrace the specificity of rivers
in the universe of fluvial systems with respect, for example, to the transport regimes or
efficiency metapopulations Gomez and Soar (2022) and Gomez (2022) identified. However,
many of the rivers in the collection of data the authors draw on are referred to in the three
aforementioned papers, and it should be abundantly clear that applying any machine learning
technique to such a profusion of data is unlikely to yield results that "lay[s] the foundations for
efficient and accurate predictions of river bedload".

With respect to the reviewer, our results demonstrate that the trained ANN model does
indeed describe the measured data well and improves upon previous application of ML-
techniques to bedload transport (See response to Reviewer 3 for more details). The
trained ANN-model agrees well with the measured dataset even without an explicit
classification of the data into "transport regimes" or "metapopulations".  That the ANN
is agnostic to these specifics is a strength of the approach - as it does not require
additional user supervision or any prior specification or sorting of the data. The
aggregated results of the ANN model are presented in Figure 2, and demonstrates an
efficient and accurate prediction relative to other available approaches. We expand
upon this point below.

This is because, as Gomez and Church (1989) demonstrated, not all bedload transport data are
created equal; some characterize 'equilibrium conditions', while other data, for example,
represent the condition of partial transport where not all size fractions are transport in the
same proportion as they are present on the bed. In light of the above it would seem, at the very
least, appropriate to specify the transport conditions to which a particular formula pertain,
because most formulae attempt to predict the maximum rate of bedload transport for a given
set of conditions and use data that match these criteria in any test of their performance.

We again acknowledge that the measured sediment load might be associated with
different (time) varying parameters such as changes in armor layer or the presence of
bedforms.

To reiterate, although measured sediment loads may incorporate reach-scale variability in factors that the model does not explicitly account for, like sediment supply/availability, such variations are likely to be embedded within a large training dataset such as this. For instance, bed grain size distribution incorporates information about the hiding effects, bedforms, and bed armoring. This information coupled with the river discharge, bed slope, and width is also likely to capture some information regarding relative sediment supply implicitly. In another example, one can argue that some combination of sediment size, river discharge, and the flow width, might carry some information about type of the hydraulics (normal, transitional, critical, subcritical, etc.), river depth, and flow resistance which are eventually related to sediment flux. In short, we expect that many of these variations in environmental conditions are embedded in the training of the ANN.

The proposed model seeks an answer to the question of "what are feasible tools that can be used for the quantification of river bedload when such data is needed?" One approach to answer that question is to measure bedload, given all the constraints, variations, and limitations, and then to develop sediment rating curves based on those data. In contrast, we propose a feasible approach that leverages existing and publicly available datasets to quantify river bedload beyond the specific sites where those data were collected. We do not claim that the trained ANN model can explicitly account for all river conditions, variations, and instances, but rather, represents one example of a simple versatile model that can predict bedload well within the proposed range of the dataset that was used to train it. We cannot explicitly determine the importance of supply/sediment availability factors at each site as these are rarely measured in a quantitative manner, however the strength of the ANN approach is in the quality of its prediction (Figures 2, 3) and either these factors are embedded within correlations with other variables or they are potentially absent and the ANN captures an expected average behavior.

Moreover, there are also obvious errors in the data set the authors use that could easily be resolved with reference to the original works. As an example, I offer my (Gomez, 1983) data that are incorrectly recorded as having a uniform D50 of 11m when, in fact, reference to the original paper shows a D50 of from ~3 mm to ~1.3 mm was specified for each measurement (transect – for reference, the size of the H-S sampler is also incorrectly specified and only an unspecified portion of the data are utilized). Moreover, the conditions these (my) data describe are clearly not a test of any bedload transport formulae or methodology for prediction!

Thank you for pointing this out. We checked the paper (Gomez, 1983) and it seems that the D50 that the reviewer is referring to is the D50 of the bedload. To clarify, the D50 that we use in this analysis is the reach-scale surface grain size, not the one for the bedload. We elected to use the bed surface size distribution for two reasons 1) practicality, as this data is more readily collected by researchers during routine

fieldwork between floods and thus, would allow the model to be applicable to more rivers without the requirement of direct collection and measurement of bedload and 2) these data are more often routinely reported than bedload size distributions. We will clarify this input parameter and the reasoning behind this selection in a revised version of the manuscript.

The bedloadweb website also provides two kinds of data for sediment grain size distributions which include both directly measured data and modeled data, which assumes a lognormal size distribution and estimates grain size percentiles that may have not been directly measured or reported. The table for grain sizes of Borgne d'Arolla (The river in question) is obtained from bedloadweb and shown below:

**Surface grain size**

| % thinner | Measure | Model |
|---|---|---|
| D5 | N / A | N / A |
| D10 | N / A | N / A |
| D16 | N / A | N / A |
| D25 | 2.67 | 3.79 |
| D50 | 7.74 | 11.00 |
| D75 | 19.57 | 16.09 |
| D84 | 25.87 | 19.00 |
| D90 | 30.07 | 24.70 |
| D95 | 41.93 | 38.95 |
| D100 | 64.00 | 95.00 |

In some cases, not all metrics for grain size distribution are reported, missing data from the size distribution is modeled based on the measured data and an assumption of a lognormal distribution.

After reviewing the original publication, we find that the reported riverbed D50 is consistent with what is reported in the bedload.web database. We've attached an annotated screenshot of the original figure below:

[Figure]

Figure 1. Cumulative size–frequency distributions of: (I) Surficial bed material; (II) Underlying bed material; (III) Bedload. Curves: (A) 18.08.79 (mobile surface) derived from 40 samples; (B) 23.08.79 (stable surface) derived from 149 samples; (C) 18.08.79 (stable surface) derived from 39 samples; (D) 17.08.79; (E) 18.08.79; (F) 19.08.79; (G) 23.08.79; (H) 17.08.79 derived from 136 samples; (I) 19.08.79 derived from 109 samples

We approximated the median grain size phi classes from the figure and find that they range from ~ -1.5 to -4 Phi, which is equal to ~2.8 mm to 16 mm. The average value of the three curves is ~11 mm, consistent with what is reported in the database.

To further address this comment, we rechecked all of the sediment grain data against both the website and original publications. Upon rechecking the data, we found that bedload.web sometimes provides the modeled values for size classes rather than the measured values when the data tables are downloaded directly in a .xls format. We went through and corrected our dataset to make sure that measured sediment sizes are used in all cases where they are available. During this correction, we found that the D16 in 5 rivers was not measured. In addition, 53 rivers used in this study did not have direct measured values for D16 and D90. In those cases, modeled values (estimated by the bedloadweb) were used. This resulted in some minor changes in sediment sizes in the database.

Using the corrected dataset, we re-trained and re-tested  the ANN and the additional sediment transport models that we looked at. We then redid the comparison of the ANN performance against those models. Using the updated dataset, MAE values across all four existing models and the ANN decreased by between .4% and 1.6%. MAE values for the ANN-trained model remain the lowest, equalling 16.1 g/s/m.  We will update all of the figures and quantitative values in the manuscript accordingly.

We will also expand the methods to better explain these aspects of the database with regard to the measured vs. modeled grain sizes and clarify when modeled grain sizes are used.

·        There is also nonconformity in the conditions the data seek to describe, in as much as some of the essential variables (the energy slope is the most common example) are reported as constant values, whereas in actuality they vary. The median grain size of the bedload is another value which is, in fact (see Gomez, 2022), rarely specified reported for each rate and is often a surrogate value (such as the median diameter of the local bed material, from which the bedload is assumed to have been derived). For all the above reasons, it would seem unlikely that any machine learning process would provide intelligible or meaningful results when applied indiscriminately to the growing wealth of bedload transport observations.

Again, with all due respect to the reviewer, we have demonstrated that the trained ANN model, which utilizes bed slope and bed surface material size distributions, reliability estimates bedload transport rates when compared to directly measured values.  We acknowledge that these variables may not capture the full complexity of the system, but these measurements are commonly reported and are feasible for other users to make.  The intent of this contribution is to take publicly available datasets, train and test a reliable and proven machine learning model (the ANN model), and assess its utility in predicting bedload transport rates on unseen data.  The result of the paper shows that the ANN is able to do this using the available input parameters within an acceptable margin of error that is either comparable to or less than predictions made using other bedload transport formulae commonly used by the fluvial geomorphology community.  In our view, accurate prediction of this unseen data by the trained ANN is both "intelligible" and "meaningful".

This is further highlighted by the site-specific model performance as explored in Figure 3.  The rating curves generated using the ANN clearly fall within the observational data for a range of parameter combinations.

Again, as previously mentioned in this response, there is a general understanding that big data incorporates many relationships between measured variables, even when these relationships may not be explicitly reported or measured directly. We agree that including more data such as sediment size distribution, or energy grade line in the ANN training may increase the accuracy of the model. However, such data are not available for the majority of existing datasets and thus, if this data were required, the total parameter space and number of samples used for training would be significantly reduced. Further, the future application of the model would then require all of this additional data to be collected for any further use.

While more data types are preferred for developing ML-based bedload model, the limitation in such data does not prohibit us from evaluating an ANN trained on existing data  in order to more accurately predict bedload transport rates. We have shown in this paper that despite any limitations in the existing data, the ANN model can predict the bedload for unseen data within a range of one order of magnitude and with a mean absolute error of 16.1 g/s/m. The proposed model is the first model that integrates such a large database towards prediction of bedload flux. Of course, the model can be further

tuned and enhanced once other types of data become widely available, but this is beyond the scope of this paper. We will expand on the balance of adding additional input parameters to train such a model versus the breadth of the application of any such model in the discussion.

· I also have a concern about the data screening, which apparently involved arbitrarily removing extreme (the upper and lower 5th percentiles) data, because "outliers generally degrade the overall performance of the resulting model", and the indiscriminate use of four bedload transport models to compare to and build intuition for the predictions of the ANN model.

Thank you for your comment. As mentioned in the paper, one limitation of a ML approach is that most data-driven models are sensitive to the presence of outliers. This issue is an acknowledged limitation of this approach and the handling the outliers by screening the uppermost and lowermost percentiles is a common practice (Dovoedo & Chakraborti, 2013; Kennedy et al., 1992). We would like to highlight that the proposed model is suggested for use within the range of the data used in the training process only, which does not include these outliers. While the model is well-suited for interpolation, we do not suggest any extrapolation for predicting bedload outside of the training range. We will clarify this further in a revised version of this manuscript.

· The authors "compared bedload flux measurements to predictions from these four bedload transport models and the trained ANN model". Do they really expect, for example, the Wilcock and Crowe or Recking models to be applicable to Mountain Creek, a sand-bed stream with an assumed bedload D50 of 0.9 mm?

We compare the results of the ANN-trained model to widely-used existing models for bedload transport that span a range of model complexity and number of input parameters. We choose to carry out this comparison to place the performance of the ANN-trained model in the context of the current state of knowledge within the field. Further, each of these models has commonly been used to estimate bedload transport rates (e.g. Millares et al., 2014; Huang et al., 2014).

What this comparison demonstrates is that the trained ANN-model, on average, clearly outperforms other existing models using the data available through the bedloadweb database and only the available data. The results of this comparison indicate that when one is limited to these specific input parameters, the trained ANN is more reliable across the full parameter space of the bedloadweb database than commonly accepted alternative models applied without any additional user knowledge or assumptions. Within the manuscript we readily acknowledge that the physics-based bedload transport equations could likely be calibrated to fit the available data as many of the equation coefficients are in practice tunable to the data at hand. However, the need for site-specific sediment flux results as calibration data severely limits the application of

these equations to practical settings as the accurate measurement of bedload transport remains a challenging and time-consuming endeavor. We believe that this comparison is important to place the strengths of this model and these results in the context of the existing state of knowledge in the field.

All previous models are valid for a mixture of sand and gravel. For the case of Mountain Creek, we'd like to clarify again that the input parameter is the surface grain size, which is reported as gravel, not the bedload grain size.  The data from the bedloadweb indicates that the bed contains both sand and gravel (D90= 2.21mm and D95 = 3.48mm). In addition, estimated bedload grain size distribution also includes gravel D90=2.12mm and D95= 3.23 mm.

Further, the site-specific MAEs computed for Mountain Creek across the ANN and the four additional bedload transport relations indicate that this site is by no means an outlier compared to the other sites analyzed in this contribution. We looked at the site-specific MAE for the Mountain Creek observations versus the ANN and four other models to make sure that the errors were not disproportionate to the other sites in the database.  This is not the case - in all but one case, the Mountain Creek MAE is within the interquartile range of all other site-specific MAEs. This indicates that the application of these models to Mountain Creek is consistent with the rest of the field sites that we looked at, even though it is a mixed sand/gravel bed river.

| Model | Site-specific MAE (g/s/m) | Percentile compared to all sites |
|---|---|---|
| ANN | 9.96 | 58% |
| Einstein | 18.88 | 55% |
| Recking | 15.98 | 40% |
| Wong and Parker | 27.24 | 23% |
| Wilcock and Crowe | 3841.5 | 40% |

Interestingly, the "worst-performing" field site across the four existing bedload transport relationships is consistently the "Rio Cordon 94-2002" dataset (Mao and Lenzi, 2007). There are two Rio Cordon records in the database, this particular record has a static width measurement, whereas the second Rio Cordon record has a dynamic width input which increases with increasing discharge. However, the differences in width with increasing discharge are <1 m, and in both records, sediment transport rates are consistently overpredicted by an order of magnitude by the four existing models.  As such, even for the Rio Cordon record with the dynamic width input parameter, MAEs exceed the 95th percentile in all cases. Given this, we suggest that there may be additional site-specific contributions to model error at this site, as many other sites in

the database have a static width input and models do not perform as poorly against observations.

Additional sources of error likely arise due to partial transport conditions and sediment supply. As Mao and Lenzi (2007) note in their publication, equal mobility conditions are only achieved during extremely high magnitude flows (RI>50 years, Q>10.42 m3/s). These conditions are not met in the "Rio Cordon 94-2022" dataset so it is likely that the transport data represent a partial mobility condition. Further, the Rio Cordon is considered to have a "moderate" sediment supply (Recking 2012). Encouragingly, the MAE for the ANN = 53.04 g/s/m. While this value is still on the "high end" of the range of errors for the ANN model, it is significantly less than the four other examples (which all overpredict observations by more than an order of magnitude). This indicates to us even more clearly the potential of the ANN for capturing more complex sediment transport behavior, including partial or selective transport or elevated sediment supply conditions, in gravel bed rivers from the 7 provided input parameters alone. It is worth stressing here that the ANN requires minimal user supervision or prior classification of the data and no site-specific calibration.

We will add these details to the results and discussion of the paper to better demonstrate how the application of the ANN model may represent an improvement from previous bedload transport models.

· In summary, I am sensitive to the fact that the authors may have been unaware of, or chose to ignore, papers that were published as their manuscript was being prepared for publication.

Thank you for your comment. We were unaware of the mentioned papers. However, to suggest that we would willfully ignore such works does not provide us the benefit of the doubt and is, quite frankly, unnecessary to suggest within the context of a manuscript review.

We thank the reviewer for their time, however we'd like to emphasize that the critical differences raised by the reviewer are largely philosophical and ignore the strength of the results of the study described within this manuscript.

However, I cannot support the arbitrary and indiscriminate use of data and the application of bedload transport models in their study. There are a number of very large (comprising hundreds to tens of thousands of measurements, see Gomez et al., 2022 and Gomez and Soar, 2022 for examples), high quality bedload transport data sets to which the bedload transport models they utilize are eminently applicable that would also serve as a rigorous test of the ANN model they have developed. My recommendation is that the paper be withdrawn as I do not consider it to be suitable for publication in its present form. The ANN model, which represents

the core of the paper, may have utility, but it needs more meticulous testing using the 'big data' to which it is inherently more applicable.

o  Gomez, B., 1983, Temporal variations in bedload transport rates: the effect of progressive bed armouring, Earth Surface Processes and Landforms, 8, 41 54.
o  Gomez, B., 2022, The efficiency of the river machine, Geomorphology, 410, 108271, doi:10.1016/j.geomorph.2022.108271 .
o  Gomez, B. and Church, M., 1989, An assessment of bedload sediment transport formulae for gravel bed rivers, Water Resources Research, 25, 1161-1186.
o  Gomez, B. and Soar, P.J., 2022, Bedload transport: beyond intractability, Royal Society Open Science[SH3] , 9: 211932, doi: 10.1098/rsos.211932.
o  Gomez, B., Soar, P.J. and Downs, P.W., 2022, Good vibrations: Big data impact bedload research, Earth Surface Processes and Landforms , 47, 129–142.

Testing the ML model using additional datasets may reveal the robustness of the model. However, we would like to reiterate that the test of the proposed model carried out in the current analysis uses unseen data that were not used in the initial development of the ML model. In addition, we checked the resources that the reviewer is pointing out. We did not find such data publicly available but rather available upon "reasonable" request or inaccessible due to dead hyperlinks within supplementary files. Further, the proposed model in this study is based on 7 particular inputs of river discharge, bed slope, flow width, D16, D50, D84, and D90. That means that for any further test of the model, all of these parameters are required. We are not aware of any other public database that offer such data, otherwise they can be added to the current database for both training and testing of the ML model.

As it was described earlier, the idea of using a machine learning approach for bedload is to rely on the information/knowledge embedded in large data and allow the ANN to iteratively hone in on possible relations, patterns, associations, and nonlinear behavior in the bedload by automatic data parsing. The addition of a single site is unlikely to provide a strong test of the model unless said site significantly expands the parameter space of the model training dataset.  Again, we'd like to reiterate that many of these differences, particularly regarding the application of a generalized model to predict bedload flux, appear to be largely philosophical in nature, and do not directly acknowledge the results of the ANN that we present in the manuscript.

Gilbert, G. K., 1914, The transportation of debris by running water: U.S. Geological Survey Professional Paper 86, 263

Einstein, Hans Albert. 1937. "Bed Load Transport as a Probability Problem." Ph.D., ETH Zurich.

Wilcock, P. R., and Joanna C. Crowe. 2003. "Surface-Based Transport Model for Mixed-Size Sediment." Journal of Hydraulic Engineering 129 (2): 120.

Lajeunesse, E., L. Malverti, and F. Charru. 2010. "Bed Load Transport in Turbulent Flow at the Grain Scale: Experiments and Modeling." Journal of Geophysical Research 115 (October): 16 PP. https://doi.org/201010.1029/2009JF001628.

Wilcock, P. R. (2001). Toward a practical method for estimating sediment-transport rates in gravel-bed rivers. Earth Surface Processes and Landforms, 26(13), 1395-1408.

Beer, A. R., & Turowski, J. M. (2021). From process to centuries: Upscaling field-calibrated models of fluvial bedrock erosion. Geophysical Research Letters, 48, e2021GL093415. https://doi.org/10.1029/2021GL093415

Deal, E., Venditti, J. G., Benavides, S. J., Bradley, R., Zhang, Q., Kamrin, K., & Perron, J. T. Grain shape effects in bed load sediment transport (pre-print)

Phillips, C. B., Hill, K. M., Paola, C., Singer, M. B., & Jerolmack, D. J. (2018). Effect of Flood Hydrograph Duration, Magnitude, and Shape on Bed Load Transport Dynamics. Geophysical Research Letters, 45, 8264–8271. https://doi.org/10.1029/2018GL078976

Parker, G., and Klingeman, P. C. (1982), On why gravel bed streams are paved, Water Resour. Res., 18( 5), 1409– 1423, doi:10.1029/WR018i005p01409.

Recking, A. (2010). A comparison between flume and field bed load transport data and consequences for surface-based bed load transport prediction. Water Resources Research, 46(3).

Masteller, C. C., & Finnegan, N. J. (2017). Interplay between grain protrusion and sediment entrainment in an experimental flume. Journal of Geophysical Research: Earth Surface, 122(1), 274–289. https://doi.org/10.1002/2016JF003943

Masteller, C. C., Finnegan, N. J., Turowski, J. M., Yager, E. M., & Rickenmann, D. (2019). History-Dependent Threshold for Motion Revealed by Continuous Bedload Transport Measurements in a Steep Mountain Stream. Geophysical Research Letters, 46(5), 2583–2591. https://doi.org/10.1029/2018GL081325

Yager, E. M., Turowski, J. M., Rickenmann, D., & McArdell, B. W. (2012). Sediment supply, grain protrusion, and bedload transport in mountain streams: SEDIMENT SUPPLY AND BEDLOAD TRANSPORT. Geophysical Research Letters, 39(10), n/a-n/a. https://doi.org/10.1029/2012GL051654

Gomez, B., 1983, Temporal variations in bedload transport rates: the effect of progressive bed armouring, Earth Surface Processes and Landforms, 8, 41 54.

Dovoedo, Y. H., & Chakraborti, S. (2013). Outlier detection for multivariate skew-normal data: a comparative study. Journal of Statistical Computation and Simulation, 83(4), 773-783.

Kennedy, D., Lakonishok, J., & Shaw, W. H. (1992). Accommodating Outliers and Nonlinearity in Decision Models. Journal of Accounting, Auditing & Finance, 7(2), 161–190. https://doi.org/10.1177/0148558X9200700205

Huang, H.Q., Deng, C., Nanson, G.C., Fan, B., Liu, X., Liu, T. and Ma, Y. (2014), A test of equilibrium theory and a demonstration of its practical application for predicting the morphodynamics of the Yangtze River. Earth Surf. Process. Landforms, 39: 669-675. https://doi.org/10.1002/esp.3522

Millares, M.J. Polo, A. Moñino, J. Herrero, M.A. Losada, (2014), Bedload dynamics and associated snowmelt influence in mountainous and semiarid alluvial rivers, Geomorphology, 206: 330-342. https://doi.org/10.1016/j.geomorph.2013.09.038.

Mao, L. and Lenzi, M.A. (2007), Sediment mobility and bedload transport conditions in an alpine stream. Hydrol. Process., 21: 1882-1891. https://doi.org/10.1002/hyp.6372

Recking, A., Liébault, F., Peteuil, C. and Jolimet, T. (2012), Testing bedload transport equations with consideration of time scales. Earth Surf. Process. Landforms, 37: 774-789. https://doi.org/10.1002/esp.3213

---

## Author Comment (AC4)

·       The authors use an ANN to derive a nonlinear relationship between several river hydraulic variables and sediment transport rate. This machine learning technique and many others have been extensively tested in the last twenty years in many similar studies that the authors overlooked. Thus, I do not think that this study adds something new to the existing literature and I suggest rejection of the paper.

Thank you for your feedback. We did a thorough literature review for this work which is presented in the Introduction. We would like to highlight that there are numerous studies that have used machine learning (ML) to predict sediment load for a specific site such as the work by Asheghi & Hosseini (2020). However, to our knowledge, there are three key studies (Bhattacharya et al., 2007; Kitsikoudis et al., 2014 & 2015) that presented a model for fluvial sediment transport, potentially applicable to other sites and comparable to this work. This is reflected in the paper within the introduction. The work by Kitsikoudis et al. (2015) is limited to sand bed rivers. An advantage of this manuscript is that the large number of field sites includes sand, gravel, and mixed bedded sites and is therefore broadly applicable. The work by Kitsikoudis et al. (2014) focuses on bedload transport within gravel-bed rivers, however the dataset is primarily drawn from a limited geographic region of the United States (Idaho, King et al., 2004). These data are generally of high-quality, however they occupy a limited portion of the river parameter space and tend to be steeper, coarser grained, and shallower than average which limits relations derived on these data to similar geographic locations (see Phillips & Jerolmack, 2019). A significant advancement of the current manuscript is the larger parameter space of river variables which allows for a significantly broader application of the model. Further, the work by Bhattacharya (2007) is limited to only 407 data points while the input to their model are derived parameters rather than direct measured values as noted within the manuscript which may reduce the broader applicability of the manuscript. There simply is not a large body of literature on the application of machine learning models to bedload transport and, to the best of our knowledge, it does not stretch back 20 years. There are a handful of site-specific studies which we will incorporate into a revised manuscript, however site-specific analysis is not the goal or advancement of this study.

A goal of this manuscript is not to have the most cutting-edge machine learning model, but a tool that can be used for prediction in the absence of long-term monitoring of bedload transport measurements. A significant strength of this manuscript that is absent in the previous literature is that the current ANN model achieves strong predictions using primarily static variables that describe the river reach alongside the river discharge which is monitored at a significantly larger number of sites across much of the United States and Europe where these data were measured. The ability to compute bedload transport for sites within the bounds of the testing data parameter space based on channel measurements that can be made at low flow and a hydrograph represents an important step forward for estimating bedload transport.

Asheghi, R., & Hosseini, S. A. (2020). Prediction of bed load sediments using different artificial neural network models. Frontiers of Structural and Civil Engineering, 14(2), 374–386. https://doi.org/10.1007/s11709-019-0600-0

Bhattacharya, B., Price, R. K., & Solomatine, D. P. (2007). Machine Learning Approach To Modeling Sediment Transport. Journal of Hydraulic Engineering, 133(4), 776–793. https://doi.org/10.1061/(ASCE)0733-9429(2007)133:4(440)

Kitsikoudis, V., Sidiropoulos, E., & Hrissanthou, V. (2015). Assessment of sediment transport approaches for sand-bed rivers by means of machine learning. Hydrological Sciences Journal, 60(9), 1566–1586. https://doi.org/10.1080/02626667.2014.909599

Kitsikoudis, V., Sidiropoulos, E., & Hrissanthou, V. (2014). Machine learning utilization for bed load transport in gravel-bed rivers. Water resources management, 28(11), 3727-3743.

Phillips, C. B., & Jerolmack, D. J. (2019). Bankfull transport capacity and the threshold of motion in coarse-grained rivers. Water Resources Research, 55(12), 11316-11330

---

## Author Comment (AC5)

In this short and well-written manuscript, the authors present an ANN model for predicting bedload flux based on a published dataset. Machine learning is increasingly used for modeling and predicting natural dynamics, with known strengths and limitations. Bedload is perhaps one of the more challenging processes to model given its strong dependency on highly dynamic and local variables. A number of models have recently been published that attempt to predict bedload over large scales (continental and global; see below). This paper is therefore quite timely and adds to the broader communities' efforts to better predict fluvial dynamics. The following issues should be addressed before it is accepted for publication. These are not very major issues but will likely require additional analysis.

> We thank the reviewer for their thoughtful review of this manuscript. We respond to individual comments below.

1. The observational dataset includes an unequal number of observations for each river - if the spatial variability is larger than the temporal variability this may lead to overfitting. The authors addressed that to a degree, but need to better discuss this issue. As it stands the model predicts temporal dynamics using observations from different rivers. ANN may be flexible enough to deal with this but, again, needs more discussion and maybe an additional analysis using some sort of average value for each site (regression may be more suitable in this case given the small sample size).

> We thank the reviewer for the comment. To explore whether or not particular variables had an outsized influence on the result, we performed a sensitivity analysis (as discussed in the manuscript) and computed the model error on both the training dataset and a validation dataset both for the ANN trained on all model inputs but also for each sensitivity test that we conducted. We found that the training and validation loss errors are consistent, indicating that the model is not overfit to the training data (See Lines 157-160). These relative values of MSE for training and validation remain consistent across all sensitivity tests of the model as well, again, indicating that the model is not likely to be overfit to the training data.

> We compared site-specific MAE values for the ANN model to both the IQR and the full range of observed bedload transport rates at each site. We found that, on average, MAE values are less than both the IQR and the full range of qs values. We found 11 instances where MAE > IQR and only one instance where MAE exceeded the full range of observed values at a site, comprising less than 10% of sites in the database. However, the median number of samples in these cases was 17, relative to a median of 50 samples across all sites.

> In addition to this, we looked at functional relationships between the site-specific model MAE for the test data versus the total number of samples at each site. We did this to ascertain whether or not the model was biased towards differences in sample size. We did not find any systematic or significant relationships between the sample size at any individual site and the computed errors between the ANN output and our test data. We

can include this in addition to the existing supplementary material in Figure S4 (referenced below).

Across 134 distinct datasets, the median number of samples is n=50. The 25th percentile for sample size is n=18 and the 75th percentile is n=83, with 82% of the data within one order of magnitude.  Only 22 sites have more than 100 samples. The largest dataset is from Goodwin Creek, has 307 samples and comprises <4% of the full database. Given this, we do not expect that any individual dataset should dominate model training.  This is further confirmed by the lack of any systematic relationship between model errors and sample size.

Because some of the input parameters to the ANN are dynamic (e.g. discharge, width), we also explored the absolute error between every individual observation in our database and the model input parameters. This is included in the supplement in Figure S4.  We find that there is no systematic or significant relationship between the absolute error across all data points and any particular input parameter.  We do find that the lowest measured transport rates result in increased errors at some sites, which is consistent with most bedload flux models in the partial or intermittent transport regime very close to the threshold for motion. (Figure S4, top-left)

In the revised version of the manuscript, we will make these details more explicit in the assessment of model performance.

2. The removal of outliers is overall acceptable but can be very problematic when using a fluvial dataset as the 'extreme' values are often just the few large rivers in a dataset. The authors warn the reader to only use/interpret the results within the range of the variables but they should more carefully examine the outliers and try to include realistic observations and maximize the dataset (and thus model) representation of large rivers.

We note that our screening of extreme values does not reduce the number of field sites, rather it excludes the most extreme values across the field sites. This keeps the larger rivers within the dataset. Within a revised manuscript we will highlight the parameter space where the model is applicable and clarify that this screening process does not reduce the number of large rivers from the original dataset.

3. The metrics selected for representing the models' accuracy are reasonable but need some justification. Why MSE and not RMSE or PBIAS or R2?

It is worth clarifying to the reviewer that we are using different error metrics to explore different things. MSE is used specifically in model training as the model iterates towards the optimized weighting of all input parameters.  MSE is the most commonly used error metric for loss functions such as those shown in Figure 1B because it penalizes larger errors moreso than RMSE, which is the square root of MSE, or MAE, which reduces the impact of these outliers. This penalization of large errors by MSE is particularly helpful in the optimization of the ANN across multiple epochs. We can include this detail in a

revised version of the manuscript in the methods section as it pertains to the training of the algorithm.

In contrast, to compare the average performance of both the ANN test data (unseen data) and the additional 4 bedload transport models, we chose to compute the Mean Absolute Error instead of MAE. We choose MAE in this case precisely because it is less sensitive to any individual outlier or large error. Because bedload transport is particularly noisy, we deemed MAE to be the most appropriate error metric to assess the average performance of each model. Additionally, as indicated by Figure 2, the existing models for bedload flux that we compare the trained ANN sometimes result in large differences between the estimated and observed values for qs. Given this, we felt that the MAE served as the most conservative comparison between the ANN and these models. We applied the same reasoning when looking at site-specific errors between the ANN and observations. We also computed the RMSE, which is provided in the supplement, and while the values differ, the relative performance of the models is the same using the RMSE metric versus the MAE (see supplementary figures). We can add a sentence about this in the main text.

We did note our choice of using MAE for the model/observation comparisons in the first paragraph of the discussion (See Lines 175-176). We feel as though this acknowledgement is sufficient for what is a fairly standard metric for quantification of errors.

4. The paper falls short in providing tools and guidelines for applying its outcomes. The paper's main outcome is to demonstrate the potential usefulness of ANN for modeling bedload flux. How can the reader use this knowledge moving forward? Will they have to develop their own ANN based on the dataset? How can it be used for other locations (as the authors suggested)? This is a common issue with ML modeling, but the authors can mitigate it with additional descriptions and tools (e.g. scripts).

We are happy to provide a documented Jupyter notebook with the ANN-model and a short user-guide in an updated supplement.

5. The authors are encouraged to explore recently published papers such as:
Cohen, S., Syvitski, J., Ashely, T., Lammers, R., Fekete, B., & Li, H. Y. (2022). Spatial Trends and Drivers of Bedload and Suspended Sediment Fluxes in Global Rivers. Water Resources Research, e2021WR031583.
Gomez, B., & Soar, P. J. (2022). Bedload transport: beyond intractability. Royal Society Open Science, 9(3), 211932.

Lammers, R. W., & Bledsoe, B. P. (2018). Parsimonious sediment transport equations based on Bagnold's stream power approach. Earth Surface Processes and Landforms, 43(1), 242-258.

Li, H. Y., Tan, Z., Ma, H., Zhu, Z., Abeshu, G. W., Zhu, S., ... & Leung, L. R. (2022). A new large-scale suspended sediment model and its application over the United States. Hydrology and Earth System Sciences, 26(3), 665-688.

Tan, Z., Leung, L. R., Li, H. Y., & Cohen, S. (2022). Representing global soil erosion and sediment flux in Earth System Models.

Thank you for the suggestion. We will review and include these papers appropriately in the revision of our manuscript.

---

## Author Response (AR1)

**Summary of author's response:**

In response, to **reviewers 1 and 3** comments related to the novelty of using ANN and previous work using ANN to predict river bedload, we expanded the literature review to more clearly articulated the reasoning behind the choice of an ANN approach and previous applications of ANN to bedload transport.  (Lines 115-175).

In response to **reviewer 2**, we expanded the introduction to more thoroughly review previous efforts to reliably predict bedload transport (Lines 15-70).  We also emphasize then known sources of variability related to bedload transport measurements, including but not limited to sediment supply, variability in erosion thresholds, and spatially variable flow (Lines 45-65). Additionally, we cite the reviewer's suggested publications where appropriate throughout the manuscript.

We corrected the grain size input data from BedloadWeb and re-trained and tested the ANN, updating all figures and values associated with model output, where those values changed (throughout the manuscript). We also provided additional information regarding the nature of the model input data (e.g. grain size fractions from direct measurements vs. interpolation methods) in Lines 190-200.   We also provide additional details related to the preparation and screening processes for the test data (Lines 210-215).

We've added text to clarify that the aim of inter-model comparison is to demonstrate the utility of ANN performance when no site-specific calibration data is available. We clarify that our aim is only to place the ANN performance into context rather than evaluate any existing model (Lines 250-260).

We also expand our discussion of the robust performance of the trained ANN relative to other models in the context of the known variability in bedload transport rates both in space and time, a concern raised by **reviewer 2** (Lines 635-645).  Rather than focusing specifically on sediment supply (the main source of variability as cited by the reviewer) we broaden this discussion to encompass any potential source of variability as also described in our expanded introduction (Lines 45-65)

To address questions from **reviewer 4** regarding the potential for model overfitting, we expanded the error analysis that we performed and describe the range of ANN model errors relative to measured variability in the original dataset (Lines 340-365). We also incorporated additional error analysis for all presented models to better weight under and over-predictions by different approaches (throughout results).

We also added the suggested citations from **reviewer 4** throughout the manuscript where relevant.  Further, following the suggestion of the reviewer we have posted the trained model and a step-by-step introductory script on Zenodo under a GNU General Public License at https://zenodo.org/record/7641313#.Y-vfbezMIeY.

---

## Referee Report (RR1)

[referee-annotated manuscript omitted]

---

## Author Response (AR2)

We thank the review for the thoughtful comments on our updated manuscript. We respond to the general comments below. We have updated all the typographical errors as identified by the reviewer in their attached PDF within the revised text.

From "Suggestions for revision" Section
**The revised manuscript is much improved. There are, however, some lingering issues from the 1st round of reviews and several new (mostly minor) issues.**
We first, thank the reviewer for reviewing a revised version of the MS.

**See the attached annotated pdf for comments and revisions. Some of the comments within the document are fairly substantial.**
We respond to these comments below in the "attached PDF" section.

**The authors all but ignored serveral of the comments in my 1st review (Reviewer #4), most notebly:**
**2. The removal of outliers is overall acceptable but can be very problematic when using a fluvial dataset as the 'extreme' values are often just the few large rivers in a dataset. The authors warn the reader to only use/interpret the results within the range of the variables but they should more carefully examine the outliers and try to include realistic observations and maximize the dataset (and thus model) representation of large rivers.**
Respectfully to the reviewer, we did include additional acknowledgement and analysis of these outliers in response to this comment in the first round of reviews. We apologize if we misunderstood the comment and the additional look at the data and the confirmation that we do not preferentially remove any sites from our training dataset did not fully respond to the reviewer's comment. To reiterate and as is stated in the paper- no sites are removed during this screening process.

To emphasize this point more thoroughly, we have added additional text acknowledging the influence of this screening process on the training dataset. We also looked at the range of sample sizes for the largest rivers in our dataset and confirmed that while the average number of samples is reduced for larger rivers, we still have a number of large river sites where the sample size exceeds the median number of samples of the entire dataset in Lines 135-145 (also pasted below in response to a PDF comment).

**3. The metrics selected for representing the models' accuracy are reasonable but need some justification. Why MSE and not RMSE or PBIAS or R2?**
We addressed the choice of MSE relative to RMSE in the response document to the reviewer's original comments. MSE is advantageous over the coefficient of determination (R2) for similar reasons, so we have added this to the existing text on line 179. There are additional reasons why R2 is less preferable than MSE, including that it is more commonly used to describe the explanatory power of a model, not the model's overall accuracy. The aim of model training is to optimize model performance around accuracy, so MSE is a natural choice. PBIAS is also less preferable to MSE because it does not necessarily capture the precision of the model, but rather the bias, or systematic distortion, of model predictions relative to data. It is our aim

during the training process to capture the overall accuracy and precision of the model, not the explanatory power of any variable or a specific distortion effect.

While we have added a reference to why MSE is preferable to R2, consistent to our description of why it is preferable to RMSE, we feel that further review of each possible type of error that we could use and its pros and cons is beyond the scope of this contribution. Given this, we chose to not add additional discussion of PBIAS which would require us to define this type of error only to say it is unsuitable. MSE is a standard choice for ML model optimization.

***Regarding the new literature I listed in my 1st review - the authors added these references as an add-on at the end of the discussion but all but ignored the research and advances they presented.***

We have added additional acknowledgement of the findings of Cohen et al. (2022) and Lammers and Bledsoe (2018) to the discussion of the model sensitivity analysis (Lines 341-342) and Lines 352-354. We have gone through the papers the authors suggested and feel as though the clearest incorporation of these works is in the discussion of potential applications of a model like the ANN presented here, given global-scale data availability. Thus, we have referenced the papers with this in mind in Lines 400-405. Discussion of the specific findings of these models regarding spatial or temporal variability in sediment flux from a global-scale modeling approach are beyond the scope of our paper's aims.

WBMsed and the MOSART-sediment model are a different class of model than the ANN presented here. The bedload transport module each model is one component of a larger modeling framework. The authors of these works have demonstrated that their models perform reliably against field observations and capture temporal and spatial variability in sediment transport well. However, referencing or explaining the specific findings of spatial or temporally variability in sediment flux on a global scale is well beyond the scope of our manuscript, which aims to present a new ANN model for bedload flux, but does not aim to explore temporal or spatial trends in ANN predictions (though certainly that could represent interesting future work!). Given this, we feel the way in which we have referenced these works is appropriate within the scope of our manuscript.

From Attached PDF

*Is this just removing flooding events in the large sites? What is the consequence for the resulting model?*

As is stated in the paper, removal of the upper and lowermost percentiles of the training dataset is a common technique to enhance model training and subsequent performance. We acknowledge that this step removes the largest flooding events of the database. However, these largest events are also likely to be the least commonly occurring, and while these more exceptional events may be interesting, this does not warrant a degradation in model performance to include them. Further, it is worth noting that the discharge range over which the trained model is reliably applied still spans many orders of magnitude and may still capture more exceptional flood or sediment transport events in smaller rivers. We have added the following to the text between Lines 135-145 to acknowledge this point more thoroughly for the reader.

*While this removal of more extreme values is an important step to ensure model quality, we acknowledge that this step preferentially removes the most extreme flow and sediment transport events from the dataset. While there is significant interest in predicting sediment transport rates for extreme flow events, these largest events are the least frequently occurring in the dataset and more data would be needed to train an ANN model to reliably predict bedload flux under these conditions. Following this screening, we maintain 134 distinct datasets, emphasizing that the training data do encompass more frequently occurring small and intermediate floods across all available sites in the database. Thus, while the trained model presented here may not be appropriate to predict bedload flux in response to exceptional events, it can still be applied over many orders of magnitude of discharge, as described above. Following this screening process, the median number of samples across all sites is n=50. For larger rivers with maximum discharges exceeding 300 $m^3$/s (n=17), the median number of samples is reduced, n=23. However, five of these larger sites do have sample sizes exceeding the median sample size n=50, with a maximum sample size of n=146 for the Mondego River* (1.8% of the full database). *Thus, following the screening process, large rivers remain adequately represented in the training dataset.*

**Since MAE is used throughout the paper, this should be explained beforehand (in the Methodology)**
We have added a new section to the methods detailing this procedure. See Lines 230-245 (described below).

**Is the MAE for all the results calculated as abs(log(O)-log(P)) ? If so, it is quite misleading.**

To clarify, we only use the log-transformed MAE when comparing the performance of each of the bedload transport models to the observed dataset. This is because model predictions can differ from observations by many orders of magnitude.

We respectfully disagree with the characterization of this method as misleading. The observational data is not normally distributed and spans multiple orders of magnitude, thus a log-transform more equally weights all orders of magnitude and portions of the existing distribution. In the calculation of MAE, MAE = sum(abs(O-P))/n where N is the Observed value, P is the predicated value, and n is the number of samples. If the predicted and observed values are within the same order of magnitude, MAE captures both over- and under-predication in an equivalent way. If the prediction over or underpredicts by over an order of magnitude, MAE becomes asymmetric and more greatly penalizes overpredictions. Based on the model performances in this contribution, this would result in model underpredictions by many many orders of magnitude to appear to perform better than models that equally over and underpredict the data by a single order of magnitude in each direction. We have added a section to the methods between Lines 230-246 to more thoroughly describing how and why we chose to calculate MAE on the log-transformed dataset. We have included this section below.

**2.3.5 Quantitative comparison of ANN performance and bedload models**
*In order to evaluate the performance of the ANN relative to these existing models, we calculated MAE for the four previous bedload transport models and the ANN model*

*based on the direct measurements of bedload flux from the BedloadWeb database within the portion of the dataset reserved for the test (n = 1,624). MAE is calculated as:*

$$MAE = \frac{\sum|observed - predicted|}{number\ of\ samples}. \quad (3)$$

*We selected MAE as the primary criteria to assess the average model performance because it is less sensitive to extreme values (Willmott & Matsuura, 2005). To better compare under and overprediction of each model across multiple orders of magnitude, we log-transformed all bedload transport observations and predictions. This is because, based on Eq. 3, predicted values that fall multiple orders of magnitude below observed values will result in very small differences between predicted and observed, which, result, by definition, in very small MAE values. In extreme cases, MAE values computed for models that, on average, underpredict the observed data by multiple orders of magnitude (e.g. Fig. 2A) can be less than MAE values for models that equally over- and underpredict the observed data within the same order of magnitude (e.g. Fig. 2d). In this case, computing MAE on log-transformed observations and model predictions more equally weights underpredictions of each model relative to model overpredictions. Further, given that the observations of bedload transport span four orders of magnitude, this procedure helps to more equally account for model errors across the full range of observations and associated predictions.*